# Data assimilation of sea surface temperature and salinity using basin-scale EOF reconstruction: a feasibility study in the NE Baltic Sea

Mihhail Zujev[1], Jüri Elken[1], Priidik Lagemaa[1]

[1]Department of Marine Systems, Tallinn University of Technology, Tallinn, EE12618, Estonia

*Correspondence to*: Jüri Elken (juri.elken@taltech.ee)

**Abstract.** The tested data assimilation (DA) method based on EOF (Empirical Orthogonal Functions) reconstruction of observations decreased centred root-mean-square difference (RMSD) of surface temperature (SST) and salinity (SSS) in reference to observations in the NE Baltic Sea by 22% and 34%, respectively, compared to the control run without DA. The method is based on the covariance estimates from long period model data. The amplitudes of the pre-calculated dominating

EOF modes are estimated from point observations using least-squares optimization; the method builds the variables on a regular grid. The study used FerryBox observations along four ship tracks from 1 May to 31 December 2015, and observations from research vessels. In the reconstruction, this data amount was compressed into daily averages over 5' N × 10' E coarse grid. Skill was tested based on daily averages on the 0.5' N × 1' E original fine grid of the model. DA with EOF reconstruction technique was found feasible for further implementation studies, since: 1) the method that works on the large-scale patterns

(mesoscale features are neglected by taking only the gravest EOF modes) improves the high-resolution model performance by comparable or even better degree than in the other published studies, 2) the method is computationally effective.

## 1 Introduction

In the coastal oceans and marginal seas, basin-scale observation, modelling and forecasting of oceanographic and biogeochemical variables is a continuing challenge. As an example from the Baltic Sea, large-scale nutrient dynamics

(Andersen et al., 2017; Savchuk, 2018) controls the level of eutrophication and hypoxia, affected by nutrient loads and changing climate (Meier et al., 2019). Placke et al. (2018) have recently shown, that Baltic Sea research models that do not include data assimilation, have still notable errors of simulating sea surface salinity (SSS) and thermohaline stratification in the main basins, compared to the good accuracy of simulating sea surface temperature (SST). Similar evaluation has been obtained earlier by Golbeck et al. (2015), based on 13 operational models used routinely in the Baltic and North Seas.

Data assimilation (DA) is a key element to improve the model accuracy with respect to observations, both in the operational forecast and the reanalysis context (Martin et al., 2015; Buiza et al., 2018; Moore et al., 2019). DA methods are built upon dynamical models and they are based on some kind of minimization (minimum variance, variational cost function formulation etc.) of modelling errors (Carrassi et al., 2018), using estimated statistical characteristics of the studied variables. Most of the

widespread methods (optimal interpolation, 3DVar, 4DVar, various options of the Kalman filter, including their ensemble formulations) use covariance as the basic statistical characteristic. Recent overviews on different DA applications in the Baltic Sea can be found in the papers by Liu and Fu (2018), Zujev and Elken (2018), Goodliff et al. (2019), She et al. (2020). Whereas there are several results from Baltic Sea reanalysis studies available (Axell and Liu, 2016; Liu et al., 2017), the operational forecasts within CMEMS (Copernicus Marine Environment Monitoring Service) do not presently include DA (Huess, 2020).

Results of DA based forecasting depend heavily on the spatio-temporal configuration of the observing system (LeTraon et al., 2019). Unlike the regular weather observing networks, observation systems in marginal seas are rather fragmented, where areas and periods of dense sampling can be neighboured by large observation gaps. Therefore, special OSE (Observing System Experiment) studies have been initiated, to find optimal observation network configurations to achieve best skill of DA (Fuji
et al., 2019). However, most of the observations of the Baltic Sea surface variables, not yet detectable by remote sensing (like salinity, nutrients etc.), stem from the FerryBox systems installed on board regularly cruising commercial passenger or cargo ships (She, 2018), and planning can be done only within the existing routes. Therefore, development of improved gap-filling techniques is a challenge.

Recently, a novel method for EOF reconstruction of gridded SST and SSS fields, using the data from (mostly) irregular and (often) sparse observations was presented as an idea (Elken et al., 2018), and then it was thoroughly tested in the NE Baltic Sea (Elken et al., 2019). The method relies on the estimate of covariance matrix from the long-period model data, which is decomposed into the full set of EOF modes. The mode values at observation points, together with the observed values, enable least-squares estimation of observational amplitudes. The method is able to follow on the regular grid the pointwise observed
temporal changes of the mean state and of the major basin-scale gradients. The aim of the present study is to implement this statistical reconstruction technique into the data assimilation of the forecast model, and to study the feasibility of such assimilation method.

The paper is organized as follows. In the section of data and methods, firstly, an overview of sub-regional oceanographic
background and short model description are presented. Observational in situ data have been compiled from three sources, and they contain shipborne monitoring and FerryBox platforms. The reconstruction method is presented in detail, and the section ends with the description of the used data assimilation method. The results section starts with the presentation of experiments in order to find the optimized parameters for reconstruction of gridded fields. The rest of the section is devoted to the analysis of the results of data assimilation experiments, ending with the skill evaluation. Finally, discussion and conclusions are
presented.

## 2 Data and methods

### 2.1 Study area and the circulation model

We have chosen the study area in the NE Baltic within 56.983°–60.65° N, 21.633°–30.3° E (Fig. 1), motivated by several Estonian national interests within the operational forecast of sea state and assessments of the marine environment. The region covers the Gulf of Finland, the Gulf of Riga and part of the Baltic Proper adjacent to these gulfs. The region is rather shallow: the mean and maximum depths are 26 m and 62 m in the Gulf of Riga (Yurkovskis et al., 1993) and 37 m and 123 m in the Gulf of Finland (Alenius et al., 1998), respectively.

The region lies in the temperate climatic zone. During the summer, yearly maximum SST exceeds usually 15 °C in July or August (Alenius et al., 1998), with highest values up to 25 °C observed in some years in the shallow coastal zones (Stramska and Białogrodzka, 2015). The warm upper layer of 10–20 m thickness is well mixed down to the thermocline or bottom, whichever of them is shallower. Occasionally, wind-driven coastal upwelling processes disrupt this warm layer (Uiboupin and Laanemets, 2009). Nearly every winter, sea ice forms with variable extent and thickness; during severe winters, the Gulf of Finland and the Gulf of Riga are fully ice-covered (Jevrejeva et al., 2004). The region is impacted by large rivers: the Gulf of Finland and the Gulf of Riga together receive 34% of the total freshwater discharge to the Baltic Sea as can be calculated from the data by Johansson (2017). As a result, there is estuarine increase of SSS from east to west (Alenius et al, 1998; Yurkovskis et al., 1993), reaching 7–8 g kg$^{-1}$ in the Baltic Proper (Kõuts and Omstedt, 1993). The Gulf of Finland has a free connection to the Baltic Proper without sill or any other topographic restriction, therefore deeper more saline waters of the Baltic Proper penetrate into the Gulf of Finland and form an estuarine halocline (Liblik et al., 2013). A shallow sill of the depth of 15 m connects the Gulf of Riga with the Baltic Proper; therefore deep layers of the Gulf of Riga can receive only surface waters of the Baltic Proper (Lilover et al., 1998). The two gulfs, located in the NE Baltic, play an essential role to the dynamics of the whole Baltic Sea (Omstedt and Axell, 2003).

For the modelling, Estonian sub-regional setup (Fig. 1) of the Baltic-wide HBM model was applied with 0.5' N × 1' E resolution containing the entire Gulf of Finland, Gulf of Riga and NE portion of Baltic Proper (Lagemaa, 2012; Zujev and Elken, 2018). The model fields are three-dimensional having 455 × 529 × 30 grid cells (by latitude, longitude and depth correspondingly) with 750 088 wet-points, and 71 986 of them on the surface with a layer thickness of 3 m. At the western open boundary, the data were taken from the Baltic-wide HBM model (Huess, 2020), operated by the Copernicus Marine Environment Monitoring Service (CMEMS, https://marine.copernicus.eu/). Atmospheric forcing was provided by the Estonian implementation of HIRLAM (Männik and Merilain, 2007). HBM uses the Arakawa C-grid, and produces forecast for 16 ocean variables including temperature, salinity, current speed and ice concentration. Detailed description of the HBM model and its validation can be found by Berg and Poulsen (2012); further analysis and evaluations are given by Golbeck et al., 2015; Hernandez et al., 2015; Tuomi et al., 2018; Huess, 2020; She et al., 2020. In particular, the CMEMS Quality Information

Document (Golbeck et al., 2018) concludes that temperature forecast between the surface and about 100 m depth is one of the major strengths of the CMEMS-V4 product, below the halocline deviations of forecast from observations increase. Regarding salinity, the values are slightly underestimated and the underestimation increases with depth.

The model setup has been designed for operational forecast. For computational reasons, it was decided to keep the operational 0.5 nautical mile grid resolution and to perform shorter feasibility experiments, instead of choosing larger grid steps and making longer experiments. The model is used routinely by the Estonian Weather Service (implemented by one of the authors, Priidik Lagemaa); SST is displayed on the web page https://ilmateenistus.ee/meri/mereprognoosid/merevee-temperatuur/ and SSS is shown on the page https://ilmateenistus.ee/meri/mereprognoosid/soolsus/. In compliance and for comparability reasons with the recent study by Zujev and Elken (2018), we chose the study period from 1 May to 31 December 2015, to be used for the DA experiments. The model experiments were conducted in the framework of operational forecast, where the forcing files were downloaded daily. There were no gaps during the study period in meteodata nor in open boundary conditions nor any other input.

## 2.2 Observational data

All available SST and SSS data from three sources were compiled:

1. Copernicus Marine Environment Monitoring Service (CMEMS, https://marine.copernicus.eu/) contains among other data sources the quality-checked data set of Baltic in-situ near-real-time multiparameter observations ftp://nrt.cmems-du.eu/Core/INSITU_BAL_NRT_OBSERVATIONS_013_032/bal_multiparameter_nrt, downloaded on 24 October 2019. This data set, accessible through free of charge registration, contains in our study region data from several FerryBox systems (automatic observations made from ferries and other ships crossing the sea areas on a regular basis). There are also a number of coastal stations, but they record mainly sea level and water temperature, whereas salinity observations are missing; therefore we are not using coastal stations. In our study area and time interval, there were not any operating buoy stations, gliders or Argo floats.

2. HELCOM/ICES database contains the results from the HELCOM marine monitoring programme and is hosted by ICES (https://ocean.ices.dk/helcom, data downloaded on 22 October 2019). It includes mainly the data from shipborne monitoring stations, where SST and SSS are easily extracted.

3. National monitoring database KESE (https://kese.envir.ee/kese/listProgram.action, search for "mereseire"), contains detailed records of all variables observed under the national environmental monitoring program. The data that were downloaded on 18 October 2019, contain different data records for every environmental variable. Except for a few cases, these data are also found in the ICES/HELCOM database. Duplicate entries were avoided from the composite data set by averaging over small time and space intervals.

The largest amount of synchronous SST and SSS data originates from the FerryBox systems, accessed through the CMEMS (Table 1). There were about 330 k (thousand) initial observation points from FerryBox, distributed over a few ship lanes (Fig. 2a) with a few hundred meters resolution and from daily to a few days interval. The analysed water is strongly mixed in the surface layer by the moving ship. Typical observation depth may be considered 5 m, although variations between the ships and due to the variable shipload exist (Lips et al., 2008; Karlson et al., 2016). There were also about 370 observations from shipborne monitoring stations. Distribution of the amounts of observations in selected temporal and longitude intervals (Fig. 2b) reveals a highly irregular pattern. Most of the observations were concentrated on the Tallinn-Helsinki transect located across the Gulf of Finland between the longitudes 24.6°–25° E. FerryBox observations were missing in the Gulf of Riga and in the eastern part of the Gulf of Finland, east from 26.5° E. In the southern part of the Gulf of Riga, available data were missing during the study period.

Two sets of compressed (averaged) FerryBox data were created for further data analysis, containing mean observed values, coordinates and observation times over the selected intervals. Firstly, for the validation and skill study, daily mean spatial averages over a fine grid (0.5' N × 1' E as in the used model) cells were created, resulting in about 110 k values. Secondly, for the EOF pattern analysis and reconstruction of SST and SSS fields, daily mean spatial averages over the coarse grid (5' N × 10' E) were created. In this procedure, the initial observations were compressed on the coarse grid by roughly 25 times yielding about 13 k average values for SST and SSS. Within the temporal averaging, it was chosen not to apply any diurnal cycle correction and all the observations at different hours were averaged to the closest midnight.

For the interpretation of model and DA results, meteorological data were taken from the model forcing fields. For the occasional comparison, CMEMS remote sensing SST Level 4 (L4) data were retrieved from the service portfolio http://marine.copernicus.eu/services-portfolio/access-to-products/ as the product SST_BAL_SST_L4_NRT_OBSERVATIONS_010_007_b.

### 2.3 Reconstruction of gridded data from point observations

For the purpose of DA, we chose to use EOF reconstruction of large-scale SST and SSS fields, using the orthogonal patterns from models following the detailed outline by Elken et al. (2019). The spatio-temporal distribution of in-situ data was too irregular to use standard interpolation and filtering algorithms like the Cressman method or optimal interpolation with approximated covariance (see an example from the same region by Zujev and Elken, 2018).

The basic option of EOF reconstruction uses at each DA time step time-fixed amplitudes, encountering the observations spanning over certain time (which can be longer than DA time step) that are transferred to the fixed times by some interpolation or filtering/averaging procedure. The amplitudes are estimated using time-fixed observations by minimizing the root-mean-square-difference between the observations and the EOF reconstruction. The amplitudes at adjacent time moments are not

directly related, but in case of longer temporal filters when observations overlap on different DA time steps, indirect relations between adjacent amplitudes become evident.


Elken at al. (2019) proposed also an advanced method with time-dependent amplitudes. Within this approach, the amplitudes and their time derivatives are estimated together with observations within a selected time interval, in order to find least squares between the observations and EOF reconstruction in the observational framework.

The main steps of EOF reconstruction are the following. During the standard EOF decomposition, the orthonormal eigenvector matrix $\mathbf{E}$ (contains the spatial eigenvectors $\mathbf{e}_k$) is found from the eigenvalue problem $\mathbf{BE} = \mathbf{\Lambda E}$ , where $\mathbf{B}$ is $M \times M$ spatial covariance matrix, calculated from the $M \times N$ spatio-temporal matrix $\mathbf{X}$ of the "values of interest" by time averaging, and $\mathbf{\Lambda}$ is a diagonal matrix that contains eigenvalues $\lambda_k$. The dataset $\mathbf{X}$ contains time slices $\mathbf{x}_i$ that are spatial state vectors at time $i$. While $\mathbf{E}$ is non-dimensional, the dimensional amplitudes (or in other words, factors) of EOF decomposition are found by $\tilde{\mathbf{a}}_i = $

$\mathbf{E}^\mathbf{T} \mathbf{x}_i$, and the decomposition is reconstructed to the "values of interest" by $\mathbf{x}_i = \mathbf{E}\tilde{\mathbf{a}}_i$. Here we have used the notation $\tilde{\mathbf{a}}_i = \mathbf{\Lambda a}_i$, where $\mathbf{a}_i$ is non-dimensional amplitude. The eigenvalues $\lambda_k$ present the variance (energy) of the eigenvectors $\mathbf{e}_k$ over the whole period, the sum of all eigenvalues equals to $\sigma^2$, the variance of $\mathbf{X}$. EOF decomposition offers the possibility to keep only the most energetic modes in the reconstruction and truncate the higher modes in $\mathbf{E}$. When $L$ most energetic modes are taken into account in the sorted list of eigenvalues and -vectors, the sum from $\lambda_1$ to $\lambda_L$ presents the explained variance and

contribution of truncated modes forms the error variance. If white noise with a variance $\varepsilon^2$ is present in the decomposed data due to sub-grid scale processes and/or sampling errors, the noise variance appears only as additive to the diagonal elements of the covariance matrix. The eigenvalue problem becomes $(\mathbf{B} + \varepsilon^2 \mathbf{I})\mathbf{E} = \mathbf{\Lambda E}$, where $\mathbf{I}$ is a unity matrix. Patterns of spatial modes remain unaffected by adding the white noise, but the eigenvalues and energy share of the modes decrease according to a factor $(1 + \varepsilon^2/\sigma^2)^{-1}$. When the sum of eigenvalues of the included dominating modes is less than $\sigma^2 - \varepsilon^2$, contribution of noise is

effectively smoothed.

During EOF reconstruction from observations $\mathbf{y}_i$, the number of observations $K$ is assumedly smaller than the number of points $M$ in the spatial eigenvectors $\mathbf{e}_k$ that are determined on the model grid and evaluated from the model statistics. For the comparison with observations, the model data $\mathbf{x}_i$ are transformed to the observation points by the observation operator $\mathbf{H}_i$ by the formula $\mathbf{H}_i \hat{\mathbf{x}}_i = \mathbf{H}_i \mathbf{E} \hat{\mathbf{a}}_i$, where $\hat{\mathbf{a}}_i$ are the "observational" amplitudes. Further, the $\hat{\mathbf{a}}_i$ values should follow least-square

minimization of reconstruction error in relation to observations $\|\mathbf{y}_i - \mathbf{H}_i \mathbf{E} \hat{\mathbf{a}}_i\|^2 \Rightarrow \min$. The expressions to find observational amplitudes and reconstructed fields are

$$\hat{\mathbf{a}}_i = \left(\mathbf{E}^\mathbf{T} \mathbf{H}_i^\mathbf{T} \mathbf{H}_i \mathbf{E}\right)^{-1} \mathbf{E}^\mathbf{T} \mathbf{H}_i^\mathbf{T} \mathbf{y}_i \, , \qquad \hat{\mathbf{x}}_i = \mathbf{E}\hat{\mathbf{a}}_i. \tag{1}$$

In the reconstruction by Eq. (1), the critical point is a possibility of spurious amplitudes based on few and unfavourable spaced observation points. Experiments with pseudo-observations (Elken et al., 2019) revealed that the values of $\hat{\mathbf{a}}_i$ of dominating $L$ modes should match the limits derived from statistics of $\tilde{\mathbf{a}}_i$, whereas higher modes with outlying amplitudes should be neglected.

Most of the oceanographic observations are not made at the same time. It may take several days or even weeks to cover a larger sea area with shipborne monitoring. When $P$ observations $\mathbf{y}_p$ are taken at different times $p$, then construct an observation operator $\hat{\mathbf{H}}_p$ that allows pointwise comparison of $\mathbf{y}_p$ and $\hat{\mathbf{H}}_p \mathbf{x}_i$ converted from gridded values at specified time $i$. Assume that within the short time span the amplitudes depend linearly on time and introduce $\hat{\mathbf{b}}_p = \hat{\mathbf{a}}_i + \mathbf{d}_i \cdot \delta t_p$, where $\hat{\mathbf{a}}_i$ is the time-fixed amplitude, $\mathbf{d}_i$ is the rate of change vector and $\delta t_p = t_p - t_i$ is the difference between the observation and reference times. The

function to be minimized regarding reconstruction errors is $Q = \left\| \mathbf{y}_p - \hat{\mathbf{H}}_p \mathbf{E} \hat{\mathbf{b}}_p \right\|^2 = \left\| \mathbf{y}_p - \hat{\mathbf{H}}_p \mathbf{E} \left( \hat{\mathbf{a}}_i + \mathbf{d}_i \cdot \delta t_p \right) \right\|^2$, which yields a system of $2L$ linear equations obtained from $\partial Q / \partial \hat{a}_l = 0$, $\partial Q / \partial d_l = 0$, $l = 1 \dots L$

$$\mathbf{Gz} = \mathbf{w}, \qquad G_{mn} = \sum_{p=1}^{P} f_m^p f_n^p, \qquad w_n = \sum_{p=1}^{P} y_p f_n^p. \tag{2}$$

Here the vector of unknowns combines the amplitudes and their rates of change $\mathbf{z} = \{\hat{a}_1 \dots \hat{a}_L, d_1 \dots d_L\}$. Instead of the full set of EOF mode values, as during standard decomposition, we take the modified/interpolated mode values at observation points; then $f_m^p = \{\hat{e}_1^p \dots \hat{e}_L^p, \hat{e}_1^p \delta t_p \dots \hat{e}_L^p \delta t_p\}$. We note that when all observations have the same time stamp and $\delta t_p = 0$, the Eq. (2) is reduced to (1).

Time-dependent reconstruction allows selecting the reference time and length of time interval. As with the time-fixed reconstruction, the highest "usable" mode is determined by checking the amplitude values with statistical limits. The method also allows estimation of amplitudes and making reconstruction by only backward observational data. This feature makes the method useful in operational forecasts, where only past observations can be taken into account for drawing the present nowcast maps.

**2.4 Method for data assimilation**

Many DA techniques use (irregular) point observations of a variable $\psi$ as the input source. In our approach, gridded maps $\psi^o$ are used; they are optimized by EOF reconstruction as described in Sect. 2.3. Therefore, in the continuous equivalent, DA is performed by Newtonian relaxation (e.g. Holland and Malanotte-Rizzoli, 1989)

$$\partial \psi / \partial t = F(\psi) - \frac{1}{\tau} (\psi - \psi^o), \tag{3}$$

which discrete form has been applied for DA, for example, using gridded climate data (Moore and Reason, 1993) or using optimal interpolation of daily satellite-based SST data (Ravichandran et al., 2013). Equation (3) is then written for DA time step $\Delta t$ in two stages as


$$\psi^f = \psi^{a-1} + \Delta t\, F(\psi^{a-1})\,, \quad \psi^a = (1-\alpha)\psi^f + \alpha\psi^o, \tag{4}$$

where $\psi^f$ is the raw forecast field calculated from the previous analysis field $\psi^{a-1}$ using only the model operator $F$ without DA during this time step, and $\psi^a$ is the new analysis field. Equation (3) contains adjustable relaxation time $\tau$ that is

transformed in Eq. (4) to non-dimensional $\alpha = \Delta t/\tau$. This is the main DA calibration parameter, since extensive use of covariance statistics, including the effects of observation errors, has been included in the estimation of gridded reconstruction of point observations. Newtonian relaxation of gridded observations, applied during the model run at DA time steps is named also "analysis nudging" (e.g. Stauffer and Seaman, 1990), which has recent meteorological applications (Bullock et al., 2018).

In practical calculations, SST and SSS observational data were reconstructed on the coarser 5' N × 10' E grid and interpolated/extrapolated by bilinear procedure to the finer 0.5' N × 1' E model grid. Such simple transition of data from coarse to finer grid includes smoothing, since $\psi^o$ lacks the details that are present on the finer grid. We have tested that the effect of added smoothing is smaller than the physical diffusion. In our study area, generation of meso- and small-scale features is of high intensity; therefore relaxation to the smooth observation fields does not apparently damp the fine grid variability. The

approach of using two grids with different resolutions is justified by irregular distribution of observations; reliable estimation is possible only for large-scale patterns of SST and SSS fields; the computationally more efficient coarser grid resolves these patterns with enough details.

The DA method is based on the full covariance matrix of irregular pattern, calculated from model results over a sufficiently

long period. Covariance is further treated using EOF modes. For the reconstruction procedure, we keep the lowest EOF modes without any approximation, covariance from higher modes is truncated. The large-scale features of the EOF reconstruction and associated DA exclude the possibility of creating spurious "bull-eye" patterns around observation points, that may happen for instance during unfavourable selection of optimal interpolation parameters. Subsequently, our DA method handles the large-scale features and excludes the possibility to assimilate smaller scale features, which can be described by the higher

modes. The method of time-dependent amplitudes is able to encounter temporally distributed observations, when estimation of linear rate of change of the EOF amplitudes over the selected interval makes sense. Mesoscale deviations from basin-scale EOF patterns follow well-defined covariance decay with space lag; therefore, they could be treated by optimal interpolation with approximated covariance or similar methods (Elken et al., 2018).

The above DA method is computationally efficient. The EOF modes are calculated prior to DA cycles. For each DA time step, only one system of linear equations of rank of the number of EOF modes (about 3-6) has to be solved for the entire grid. The coefficients of the matrix are found by summation of the products of EOF mode values over the observation points (Eq. 2). For comparison, optimal interpolation requires solving the system of linear equations of rank of the number of observation points (about 100) for each grid cell (about 1000), with a single inverse matrix calculated for the time step.


The model skill with respect to observations was evaluated over those grid cells - time span pairs when observations were available. Since observations covered only a small part of the study domain, DA results were also compared with control run without DA, but then it is possible to only analyse the changes due to DA, without conclusion of possible improvement. Standard statistical characteristics were calculated for individual fields: mean, standard deviation, in case of differences (for

example, relative to observations): bias, RMSD (centred root-mean-square difference that equals to the standard deviation of difference field), and the Pearson correlation.

## 3 Results

### 3.1 Experiments on EOF reconstruction

#### 3.1.1 Covariance, modes and reconstruction tests

The EOF modes were calculated on the coarse grid (5' N × 10' E) on the basis of space-averaged results from the fine grid (0.5' N × 1' E) model, running from 1 July 2010 to 30 June 30 2015 (Elken et al., 2019). This analysis revealed that mean distributions of modelled SST and SSS, serving as the basis for calculation of deviations in the variability studies, were close to the climatological maps calculated on the basis of observations (Janssen et al., 1999). Highest temporal variability was

found in the shallow coastal areas for SST, whereas largest SSS variations were revealed near the larger river mouths and in the NE area of the Gulf of Finland. While temporal changes strongly dominate in the variability of SST, spatial changes prevail in SSS variability.

Calculated SST and SSS covariance matrices have significant spreading of individual values over pairs of points, especially

for the dominating gravest modes where big covariance values may occur over large distances. Covariance of residual fields (sum of higher EOF modes) has a decay scale about 30 km with increasing space lag, both for SST and SSS. The first, most energetic EOF modes have nearly "flat" patterns without sign change (energy share 97.6% for SST and 36.2% for SSS); their amplitudes are dominated by a seasonal signal. Space-dependent mean biharmonic seasonal cycle was not removed from the model time series prior to the analysis, since special experiments revealed only a small effect of seasonality suppression on

EOF mode patterns. Second EOF mode of SST (1.3%) presents differential heating and cooling in shallow areas, compared to the deeper offshore waters. Transverse anomaly stripes near northern or southern coasts, like due to coherent upwelling and downwelling in the region, were evident in the second SSS mode pattern (16.9%) and third SST mode pattern (0.31%). There is also a pattern of SSS changes in the freshwater spreading pathway near the northern coast of the Gulf of Finland (third SSS mode, 7.1%) and longitudinal SST changes in east-west direction (fourth SST mode, 0.14%).


The data set used in the present DA study (Fig. 2) is rather irregular, compared to the reconstruction experiments by Elken et al. (2019). Therefore, we revisit the covariance issues and perform additional reconstruction tests, before finding in the next subsection the best options for the automatic reconstruction procedure. Spatial interrelation of observed values at a specific point to the values in the rest of the region is found from the extract of the spatial covariance matrix, which can be shown as a

map. One example of SSS covariance with a frequently sampled HELCOM monitoring station BMP F3 is shown in Fig. 3. The covariance of three dominating EOF modes (Fig. 3b) comprises most of the unfiltered data covariance (Fig. 3a) at large distances. High covariance locations have clear basin-scale geographical explanations: under the similar weather and seasonal forcing, which is spatially nearly uniform, SSS changes in distant river influence areas are closely interlinked. Correlation (not shown) may exceed 0.4 at distances greater than 500 km; therefore, assumptions of fast decay of correlation with space lag

(like using the Gaussian covariance approximation), adopted in offshore areas with negligible coastal influence, are not valid. Covariance of residuals to the large-scale variations are presented by higher EOF modes (Fig. 3c). Such smaller scale variations have nearly Gaussian structure, with elliptical anisotropy stretched along the axis of the basins similar to the results by Høyer and She (2007): spatial scales in Fig. 3c are 30 km and 15 km along the main axis and perpendicular to the axis, respectively. Similar regularities – physically explained high covariance at large distances, localized covariance patterns for the higher EOF

modes – were found for other points of reference, both for SSS and SST fields.

EOF reconstruction method relies on the full covariance matrix, without any approximation. Full covariance matrix can be implemented in optimal interpolation as well. While EOF method needs to limit the number of included modes, smoothing in such way smaller scale variability and observational errors, optimal interpolation needs to include observational error variance

("nugget effect" in terms of Kriging method, equivalent to optimal interpolation); otherwise the system of underlying linear equations may become close to singular and the result may become unrealistically spiky. In some examples (not shown), EOF reconstruction and optimal interpolation based on full covariance produced similar results, but these relations need further studies. When observed values were close to the model-computed climatological background, visual similarity was caused mainly by the dominance of spatial gradients of mean SSS over the spatio-temporal variability. Optimal interpolation with

Gaussian approximation to the covariance produced realistic results in the neighbourhood of observation points, but gave unrealistic patterns and values in the distant SW extrapolation area.

### 3.1.2 Finding the parameters for reconstruction of gridded fields

Multiple checks performed on our data set suggested that three gravest modes were included in the EOF reconstruction. In order to find the best options for reconstruction, experiments were made with different intervals (time window) $t_R$ around the reference time $t_i$; including the observations within time window from $t_i - t_R/2$ to $t_i + t_R/2$. The results were evaluated to fulfil the goals:

      A. Small RMSD between the observed values and the reconstructed fields;

      B. Small number of gaps in the reconstructed time series;

      C. Low number or missing presence of "spikes" and/or "jumps" in the time series.

Two basic options for temporal handling of the reconstruction procedures were tested:

(a) application of procedure by Eq. (1) of time-fixed amplitudes; time average of observations was taken for each grid cell, time adopted in each grid cell as constant reference time,

(b) full application of the procedure by Eq. (2) of time-dependent amplitudes; all the daily mean observations (average was taken also over coordinates and time) were kept separate for each coarse grid cell where the observations existed.

In addition, procedure by Eq. (2) was tested with an option with time average of observations in each grid cell, and with selection of closest to the reference time observations. These experiments provided more spikes and 70% higher RMSD than the basic options (a) and (b) and they were neglected from further consideration.

As a first step in all the experiments with variable time window, the EOF amplitudes of the mode $k$ were checked for the limit $|\hat{a}_{i,k}| < 2\sqrt{\lambda_k} = 2\sigma(\tilde{\mathbf{a}}_k)$, where $\sigma$ denotes standard deviation. DA data for the days with higher amplitudes were left blank since these reconstruction results most frequently became unrealistic. In addition, when the number of observations was less than six, reconstruction was not performed.

The time windows $t_R$ for experiments (a) and (b) were selected to be 10, 20 and 30 days. Elken et al. (2019) have found that the correlation time scales (e-fold drop, correlation value 0.368) of EOF SST amplitudes were 65 days for the seasonal 1st (overall heating/cooling) and 2nd (faster heating/cooling in shallow coastal areas) modes, and 15 days for the 3rd "upwelling" mode. Time scales of the SSS modes were 65 days for the 2nd and 3rd mode, representing the large-scale gradients, and 110 days for the 1st mode describing long-term variations of mean salinity.

Methods of time-fixed (a) and time-dependent (b) reconstructions revealed similar statistical results during the study period in 2015, whereas RMSD between observed and reconstructed values of (a) was by 5% larger than of (b). By increasing the time window, RMSD of reconstruction slightly increases due to the stronger smoothing. The smoothing effect can be seen from the

reconstruction examples given in Fig. 4. It should be noted that the reconstruction is designed to yield the best approximation to the observations over the entire region; therefore, it does not need to present the local best fit at individual points.

Network of observations, available during the study period, appeared favourable for the reconstruction, although observations
were missing in the southern part of the Gulf of Riga and eastern part of the Gulf of Finland. With a time window of 30 days, there were no reconstruction gaps identified during the study period, determined for both of the methods by the above described amplitude limit criteria. Smaller time windows yielded some gaps in 2015. During the longer period from 2010 - 2018, gaps were found in most of the years (except our study period), whereas shorter time windows result in more reconstruction gaps. Detailed comparison of the time-fixed (a) and time-dependent (b) methods revealed that time-fixed reconstruction might create
spurious "jumps" when there is a gap in observations which length is close to the time window. In that case, backward average is taken before the gap and forward average after the gap, which may result in "jumpy" results. Time-dependent reconstruction, which also accounts for the temporal changes within the time window, handled such situations more smoothly.

Based on the results from reconstruction experiments, the gridded SST and SSS data were reconstructed by the time-dependent
reconstruction method (b) using three gravest modes in the time window of 30 days, centred around the assimilation time like during reanalysis. These gridded fields were applied in the DA relaxation scheme Eqs. (3)–(4).

### 3.2 Data assimilation experiments

In DA experiments, gridded observational data were pre-calculated each day using the time-dependent EOF reconstruction method with a time window $t_R$= 30 days as presented in Sect. 3.1. Reconstructed SST and SSS fields were calculated on the
coarser five nautical miles grid and interpolated bilinearly to the fine 0.5 nautical mile grid. Further on, each day DA was made on the fine grid using the procedure Eqs. (3)–(4) with $\Delta t = 1$ day. Two basic experiments were conducted, with relaxation time 10 days (weight of observations 0.1, experiment code DA01, assimilated fields SST01 and SSS01 for temperature and salinity, respectively) and with relaxation time 5 days (weight 0.2, experiment code DA02, field codes SST02 and SSS02). In addition, a variety of short-term trials was performed in a preparatory phase (results graphically not presented) which led to the two
basic experiments. Comparison data were coded as FRT and FRS for temperature and salinity of control run without DA, and FBT and FBS for observed FerryBox data, respectively.

### 3.2.1 Example from the beginning of August

There was an interesting oceanographic situation in the beginning of August, when moderate but extensive upwelling SST pattern at the northern coasts of the basins (Fig. 5), with some effects on SSS (Fig. 6), was combined with fast heating of  thin
(6-9 m) surface layer (Fig. 7). Since the middle of July, moderate winds with speeds from 4 to 6 m/s, which had a westerly zonal component (favoring upwelling at the northern coasts of the basins), were blowing above the Gulf of Finland. After 3 August 2015 (the maps in Figs. 5 and 6 are taken on this date), wind ceased and air temperatures increased by 10 August across

the study area up to 25–27 °C in the Gulf of Finland and up to 31 °C in the southern Gulf of Riga, creating a thin layer of warm surface water. Heating of surface waters was favored by high nightly air temperatures, higher than SST. Vertical profiles (not shown) in the Gulf of Finland revealed a deep thermocline at 40 m depth near the southern (downwelling) coast and a shallower thermocline near the northern coast; the warm surface water column was near Tallinn two-three times thicker than near Helsinki. From the end of July to 10 August, warming resulted in an increase of SST (Fig. 7) near Tallinn from 16.5 °C to 18.5 °C and near Helsinki from 14.5 °C to 18 °C.

The SST maps presented in Fig. 5 include control run, reconstructed in-situ observations, one experiment with DA (the other experiment yielded similar results) and satellite observations. When warm waters with SST above 17 °C dominated the study area, all the maps revealed moderate upwelling near the northern coasts of the basins. However, the minimum temperatures and the spatial extent of the colder waters were different. Warmest "cold" waters were observed on satellite images. While satellites measure SST of a thin surface layer, then FerryBox and models acquire temperature over much thicker layer. It is known that in the Gulf of Finland satellite and FerryBox can have similar SST values in case of winds stronger than 5 m/s (Uiboupin and Laanemets, 2015); at smaller wind speeds the SST bias can be 1–3 °C in reference to FerryBox observations. Within these accuracy limitations, satellite observations presented in Fig. 5d confirm the model patterns to some extent. The control run (Fig. 5a) was characterized by too high SST contrasts, compared to the satellite data (Fig. 5d). From the earlier study by Zujev and Elken (2018), it is known that the free model without DA forecasts faster heating and cooling of shallow coastal areas and slower heat dynamics in offshore areas. Data assimilation (Fig. 5c), made using the reconstructed FerryBox data (Fig. 5b) reduced discrepancies with satellite observation. The major large-scale differences between the satellite data (Fig. 5d) and the best DA02 (Fig. 5c) can be outlined as follows: (1) the colder upwelling water extended on the satellite image further to the east, (2) warmer waters were found on the satellite images in the southern Gulf of Riga, near the Daugava river, and in the shallow areas between the Estonian islands, (3) in the Gulf of Riga, a strip of colder waters was modelled along the western coast, while satellite observations revealed warmer waters near this coast.

There were also numerous mesoscale features evident on SST (Fig. 5) and SSS (Fig. 6) maps, like colder upwelling filaments along the northern coasts of the Gulf of Finland and the Gulf of Riga, and decaying anticyclonic warm-core eddies near the southern coast of the Gulf of Finland. The Irbe Front (Lilover et al., 1998; Raudsepp and Elken, 1999), formed by the salinity difference between the Gulf of Riga and the Baltic Proper, was found by the SSS maps in the outward position, stretching from the strait towards the open sea. This salinity structure was also repeated in the SST patterns; the satellite observations confirmed the predicted outward position during the taken snapshot. The model predicted that in the Gulf of Riga the Daugava river waters were spreading by narrow coastal strips of lower salinity in both the NE and NW directions (Fig. 6).

### 3.2.2 Time series in the areas of dense observations

Locations with dense observations allow us to validate the model and visually evaluate assimilation quality. We compared SST and SSS data of control run (FR) and DA options DA01 and DA02 with FerryBox data (FB) at two points near Tallinn and Helsinki (Fig. 7). While SST followed the seasonal cycle, with weather-dependent deviations, then SSS behaviour was more irregular. In the given variation scales of SST and SSS (16 °C and 2 g kg$^{-1}$ respectively), all the compared SST data sources were more similar to each other than that of SSS. Still, most of the time the assimilation curve (DA02) was closer to

the FerryBox observations than the control run, for both SST and SSS.

Warm conditions in the beginning of August (Sect. 3.2.1) are clearly visible on SST time series (Fig. 7a, c). Comparing the values near Tallinn and Helsinki, the southern part of the Gulf of Finland was roughly 2 °C warmer than the northern part, whereas the northern part had an unstable day-to-day pattern, possibly due to the fluctuations of the upwelling pattern. This is

consistent with the spatial maps given in Fig. 5. Near the southern coast, an upwelling event occurred in September, reducing SST during a few days nearly by 4 °C (Fig. 7a). Larger SST drop during the southern coast upwelling (at easterly winds), compared to the northern coast upwelling (at westerly winds of the same magnitude), is explained by the steeper topography slopes in the southern part of the Gulf of Finland (Laanemets at al., 2009). This upwelling event was properly resolved by all the data sets, with DA02 being closest to observations. In general, a free model without DA expected warming at a lower rate

during summer and was more precise in autumn, while both assimilation experiments properly corrected the SST and SSS values. However, in some cases, assimilated temperature was somewhat higher than observed and modelled SST.

Assimilation resulted in one major SSS improvement in early summer when the model predicted upwelling with too high salinity near Helsinki. Nevertheless, in some cases DA made minor corrections at one of the locations, ignoring observations

and sticking to the control run (e.g. late July - early August near Tallinn, October near Helsinki). When the model overshoots at both locations, DA properly corrects temperature and salinity values. This implies that DA tends to better correct the mean values than the cross-gulf gradients.

In the salinity time series, a "freshwater event" with reduced salinity was observed in the Gulf of Finland at the end of

September and beginning of October. In the daily SSS data (Fig. 7b, d) the event was spiky, possibly due to the mesoscale features not assimilated in the present study: without DA, the eddies tend to have random phase, and the spikes in the time series of different model options and observations do not need to be coherent. However, in the weekly averaged data (not shown) the mesoscale activity was suppressed and the fresh event appeared simultaneously in all the data within the central and western part of the Gulf of Finland.


Regarding the assimilation experiments DA01 and DA02, there is no proportionality between the options of 5 and 10 relaxation days in terms of DA performance, as can be seen from Fig. 7. They diverged as the region experiences a temperature drop or daily trend change. Both options of assimilated SST could either coincide for a long time or go in parallel, but DA02 was systematically closer to the FerryBox observations. Salinity fluctuations had larger amplitudes in the free run without assimilation, but both DA algorithms, with a "thumb" rule - the bigger the weight, the bigger the change, had properly corrected them. Only in December DA01 showed better results, being closer to the FerryBox salinity than assimilation DA02.

### 3.2.3 Spatio-temporal dynamics

We have chosen to compare assimilation with best results (DA02) to the control run without data assimilation (FR), and track the continuous time-latitude changes of SST and SSS (Fig. 8) in two sub-basins - Gulf of Finland and Gulf of Riga along the coast-to-coast transects given in Fig. 2a. Using DA, temperature was corrected approximately by 1–2 °C, and salinity by less than 1 g kg$^{-1}$. Major systematic change (in the Gulf of Finland this was validated as improvement, see further Sect 3.2.4) was seen near the coasts and in spring/autumn periods, while summer temperatures underwent minor corrections. Salinity corrections had a more uniform distribution and smooth drifting pattern - DA consistently increased SSS values with time and southwards in both of the sub-basins.

Data assimilation had increased SST in the Gulf of Finland in open waters during the warming period and in late autumn all across the gulf, and had decreased in the coastal areas during the warming period, whereas near the northern coast this decrease continued until September. In the Gulf of Riga, SST increase dominated throughout the study period, but it was interrupted occasionally by basin-wide events when DA had decreased the temperature compared to the results from FR. Largest corrections of both SST and SSS were evident in the coastal waters. Salinity was increased by DA in most of the cases in the Gulf of Finland, except for May-July near Tallinn. Largest increase of SSS occurred in November and December, when control run results dropped compared to the earlier period.

Some unusual basin-wide events can be found on the difference charts in Fig. 9. For example, abrupt warming of the surface around 10 August 2015 (Sect. 3.2.1) was correctly predicted by the free run model (Fig. 7c), but it was over-smoothed by the data assimilation. Similar line in December on both charts denotes occurrence of fronts of cold and saline water due to strong winds and storms.

As there are not enough observations available in the Gulf of Riga for validation, we cannot definitely say whether DA improved the situation in the region and to what extent.

### 3.2.4 Evaluated skill

Ocean model performance (e.g. Stow et al., 2009; Golbeck et al., 2015; Placke et al., 2018) is usually evaluated by the differences between the observations and the model results, transferred to the times and locations of observations that they can be directly compared. The overall mean difference (over time and space) is termed bias and the standard deviation of

differences at all the observation points is denoted as RMSD (centred root-mean-square difference). The forecast skill is usually non-dimensional, with the RMSD of the studied option (in our case, DA) scaled to reference data (FR in our case).

The present ocean model has a fine 0.5' N × 1' E resolution of about 930 m (Sect. 2.1), therefore for comparison with observations we used a simplified approach and took averages of observations over the model grid cells over daily time span

(Sect. 2.2). Such compressed fine-resolution observational data set, still having about 110 k points for SST and SSS, was originating mainly from the FerryBox (FB) lines (Fig. 2), and it covered central and western parts of the Gulf of Finland and the neighbouring part of the Baltic Proper. Areas with lower salinity in the eastern Gulf of Finland and in the Gulf of Riga had only a small number of observations.

Data from the DA experiments DA01 and DA02 were compared to the same compressed observational FB data as the data from the control run without assimilation (FR). The problems of performance evaluations of operational ocean models were addressed by Hernandez et al. (2015). In our study, the option of withholding the observations was performed: it was evaluated how much the DA result will change if DA is performed using 50% of the available data (Gregg et al., 2009). The present implementation of EOF DA used about 13 k observational averages over coarse 5' N × 10' E grid. The reconstruction procedure

by Eqs. (1)–(2) has no direct connection to the ongoing modelling (although it includes statistical results from longer model runs) and the fields of $\psi^o$ in Eqs. (3)–(4) are the only link where observations enter the DA process. The experiments which took every second available observation "box" into account (this resulted in mean sampling interval along ship tracks about 20 km instead of 10 km) revealed that performing DA during the study period with reduced data set (6.5 k averaged observation data instead of 13 k) changed RMSD of SST by only 1% and of SSS by 2%, whereas the RMSD values were 0.05 °C for SST

and 0.027 g kg$^{-1}$ for SSS. It was evaluated over the full time span and domain using 182 k coarse grid cells; correlation between the data sets was higher than 0.999. We have also checked reconstruction results with FerryBox data only, excluding the data from shipborne monitoring stations. Compared with the full data set, largest (but still minor) differences with RMSD of SSS up to 0.03 g kg$^{-1}$ were found in the Gulf of Riga and the eastern Gulf of Finland, where FB data were missing. Consequently, for our large-scale approach DA results are robust to the reasonable variation of data amount and we used FB data for reference

in the skill evaluations.

Evaluated skill metrics are presented in Table 2. Only those fine grid points were used for metrics calculation, which had respective value of FerryBox observations on the same day. Wet-points of the model without corresponding observation value were left out from the procedure.


The skill properties presented in Table 2 reflect that DA improves the model performance significantly: RMSD of SST was reduced by 22% and SSS by 34%, compared to the control run. From DA01 to DA02, slight improvement of DA performance was observed, therefore we adopted DA02 as the major result. Spatial pattern of RMSD change between the DA and FR (Fig. 9) reveals that larger reduction rates (up to 50%), both for SST and SSS, were found in the observation-covered areas in the
Gulf of Finland. Too cold waters produced by FR near the northern coast of the Gulf of Finland were effectively corrected by DA (see also Fig. 5), therefore highest improvement percentage scores were detected in this region. Near the western open boundary, non-assimilated SST and SSS values of the larger model were advected into the area, therefore RMSD reduction was small, or even negative for SSS.

The applied EOF DA method does not assimilate mesoscale variability. Applying the weekly average statistics like Zujev and Elken (2018), further reduced RMSD by 13% for SST and 9% for SSS, compared to the daily data in Table 2. Weekly statistics suppresses the mesoscale variability and reveals better match between the DA and the observations. DA decreased the bias, especially for SSS. At the same time, correlation of SSS between DA and observations increased considerably. We may conclude that DA made major improvement in modelling of SSS. Still, RMSD to the observations makes 62% of observed
standard deviations, calling for further improvements in modelling SSS in the Baltic Sea. Modelling of SST is more accurate than SSS already without DA: RMSD of control run (FR) makes 18% of the standard deviation of observations for SST and 94% for SSS.

## 4 Discussion

Baltic Sea is considered as one of the most studied marine areas in the world (e.g. Andersen et al., 2017). However, the large
observational data sets are distributed unevenly. If we divide our study area into 744 eddy-averaging 5' N × 10' E grid cells, then during the study period 330 k FerryBox observations covered only 18% of the sea region. Shipborne monitoring added more 8% coverage of the area, but with much smaller frequency of sampling. Having in mind that the ocean models tend to deviate in the NE Baltic from the observations not only by constant bias but also for large-scale and longer-term response, introduction of non-local, region wide data assimilation is of high importance.


It is interesting to consider how our statistical evaluations of model and DA performance, given in Table 2, compare with other Baltic Sea studies. For remote sensing versus in situ reference, Kozlov et al. (2014) have found RMSD 1.31 °C in the Curonian Lagoon. Uiboupin and Laanemets (2015) have estimated RMSD of various satellite products to FerryBox in the Gulf of Finland

from 0.29 to 0.98 °C. Our control run gave RMSD 0.72 °C. Golbeck et al. (2015) compared SST from 13 models with satellite
data and found in the Baltic Sea yearly RMSD for SST 0.65–0.87 °C. They found larger relative spread of SSS ensemble
members than of SST: deviations in the Gulf of Finland between the models were nearly up to 1 g kg$^{-1}$, while the average SSS
is only about 4 g kg$^{-1}$. Unfortunately, there were not enough validating observations for SSS available. Fu et al. (2011) found
for the control run RMSD for SST even larger, 1.0 °C, based on satellite observations. They also used DA with ensemble
optimal interpolation and found that DA reduced RMSD between the forecasts and observations by 25% for SST and 34% for
SSS. With our simpler and less computationally demanding EOF DA technique, similar RMSD reductions have been obtained
(Sect. 3.2.4) compared to earlier studies.

We have developed and tested an EOF-based relaxation technique where the large-scale observed fields to be assimilated are
pre-calculated independently from the ongoing model. From sparse observations, it is possible to estimate the amplitudes of
only the gravest, large-scale EOF modes. The EOF DA method handles large-scale features over the sea basin(s), like change
of mean SST, SSS and their gradients, including differential heating in coastal and offshore areas, major patterns from
upwelling, and spreading of river discharge. The method can work well with irregular data, but cannot resolve mesoscale
features in the areas of dense observations, because the EOF amplitudes of higher modes get noisy, according to our
experiments. Optimal interpolation, successive corrections and similar methods assume usually localized covariance and/or
radius of influence (e.g. Axell and Liu, 2016); they work well in resolving mesoscale in dense sampling areas, but regions of
rare observations remain unaffected by DA. For mesoscale range, in our study area there are only satellite observations of
surface variables available. They were omitted from our study, since salinity as a variable of primary interest can be presently
determined in the Baltic only in situ. It is possible to implement on top of EOF DA more traditional localized DA methods to
assimilate mesoscale data when and where such data are available. Studies on using EOF DA for handling large-scale data are
also ongoing in the UK Met Office by Daniel Lea (Haines, 2018).
We have tested the EOF-based DA in centred time window of 30 days, based mainly on available FerryBox data during the
study period. As shown by reconstruction experiments by Elken et al. (2019), the time-dependent method can also work with
backward observations as if it occurs during operational forecasts. When more observations become available, for example
from new automated buoy stations, Argo floats and gliders, the time window can be shortened. Full covariance matrix
estimated from the model results is the backbone of the EOF DA method. Prior and/or complementary to implementation of
the method into operational practice, detailed covariance studies using results from multiple models could be useful, as well
as additional reconstruction and DA studies using more data sources over longer periods.

The EOF DA method has some practical advantages. Firstly, it has small computational effort compared to the localized
methods like optimal interpolation etc. Secondly, intermediate results are in the form of maps that are easily understandable
and can be checked visually or taught to be analyzed by artificial intelligence. For optimizing the observational data needs, the
concept of OSE (Observing System Experiments) that checks various data configurations for DA performance, are highly on

the agenda. Since the quality of DA and forecast are primarily determined by the quality of EOF reconstruction (when extensive mesoscale observations are not available), then it would be possible to save a significant amount of computing power and perform most of the experiments using orthogonality of the EOF basis vectors.

There are obvious possible extensions of the EOF DA method to other variables and layers: improvement of stratification modelling, extension to biogeochemical models and DA of oxygen, nitrogen and phosphorus. Applicability depends on how well the model reproduces the studied fields and their covariance, and much variance is explained by the major EOF modes. There are a number of questions that may be addressed, like: What is the minimal amount of observations needed to produce decent results? What areas are reconstructed with higher accuracy with given observation design, nearshore, offshore, open basins? What areas are most problematic to reconstruct, complicated coastline, straits and channels, semi-enclosed basins, regions of river influence? Are there some specific locations that can be used as a proxy for larger regions? Is it possible to measure SST/SSS just at these points in order to give enough input for successful reconstruction?

## 5 Conclusions

The present study was aimed to implement EOF-based statistical reconstruction technique into the data assimilation of the forecast model, and to study the feasibility of such assimilation method. Gridded EOF modes were determined from the 5-year long model results. "Observational" EOF amplitudes were found each day to minimize the RMSD between the reconstructed and observed values at the observation points, using time-dependent technique where both the amplitudes and their time rate of change were searched for the best fit. In this procedure, a time window of 30 days was selected that ensured acceptable SST and SSS reconstruction patterns by three gravest EOF modes throughout the whole study period from 1 May to 31 December 2015. The study used about 330 k (thousand) FerryBox observations along four ship tracks from 1 May to 31 December 2015, and 370 observations from research vessels. Statistically gridded observations were daily assimilated into the model by the relaxation techniques, using restoring times of 5 and 10 days.

The tested EOF-based data assimilation (DA) method decreased RMSD of surface temperature (SST) and salinity (SSS) in the NE Baltic Sea by 22% and 34%, respectively, compared to the control run without DA. Using the observation-estimated amplitudes of the pre-calculated gravest model-based EOF modes, the method is able to follow on the regular grid the pointwise observed temporal changes of the mean state and of the major basin-scale gradients. DA with EOF reconstruction technique was found feasible for further implementation studies, since: 1) the method that works on the large-scale patterns (mesoscale features are neglected by taking only the gravest EOF modes) improves the high-resolution model performance by comparable or even better degree than in the other published studies, 2) the method is computationally effective.

**Code and data availability**

The model code has been developed by the Baltic MFC partners. Presently it is frozen and not anymore developed. The DA scripts and demonstrated model results can be requested by contacting the corresponding author. All the used observational data are freely available as described in Sect. 2.2.

**Author contribution**

MZ carried out DA experiments and performed analysis of the results. JE worked on theoretical aspects and performed gridded reconstruction of observations. PL worked with the circulation model. All authors contributed in discussion, planning and writing.

**Acknowledgements**

The study was supported by the PhD program for Mihhail Zujev and the institutional research funding. A larger BAL MFC team did development of the HBM model within the EU projects MyOcean, MyOcean2 and MyOcean-FO. There is an ongoing activity to develop and maintain Baltic monitoring and forecasting services within the CMEMS. This cooperation is highly acknowledged.

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

**Table 1: FerryBox data from 1 May to 31 December 2015 in the NE Baltic used in the present study.**

| Ship | Main route | Operating institute | Number of initial observations |
|---|---|---|---|
| Baltic Queen | Tallinn – Helsinki | Marine Systems Institute, Tallinn University of Technology | 63 368 |
| FinnMaid | Helsinki (Vuosaari) – Travemünde | Finnish Environment Institute | 142 235 |
| Silja Serenade | Helsinki – Mariehamn – Stockholm | Finnish Environment Institute | 60 228 |
| Victoria | Tallinn – Mariehamn – Stockholm | Estonian Marine Institute, University of Tartu | 65 037 |

Table 2: Statistics of daily data in 0.5' N × 1' E grid cells with FerryBox (FB) observations: free model run without data assimilation (FR), data assimilation DA01 (observation weight 0.1), DA02 (weight 0.2) and FB. Bias, RMSD and correlation are taken with reference to FB.

| | FR | DA01 | DA02 | FB |
|---|---|---|---|---|
| **SST (°C)** | | | | |
| Mean | 12.03 | 12.15 | 12.25 | 12.48 |
| Standard deviation | 3.98 | 3.92 | 3.93 | 3.97 |
| Bias | -0.45 | -0.33 | -0.23 | 0 |
| RMSD | 0.72 | 0.59 | 0.56 | 0 |
| Correlation | 0.98 | 0.99 | 0.99 | 1.00 |
| **SSS (g kg$^{-1}$)** | | | | |
| Mean | 5.61 | 5.79 | 5.85 | 5.93 |
| Standard deviation | 0.35 | 0.29 | 0.31 | 0.37 |
| Bias | -0.31 | -0.14 | -0.08 | 0 |
| RMSD | 0.35 | 0.24 | 0.23 | 0 |
| Correlation | 0.52 | 0.76 | 0.78 | 1.00 |

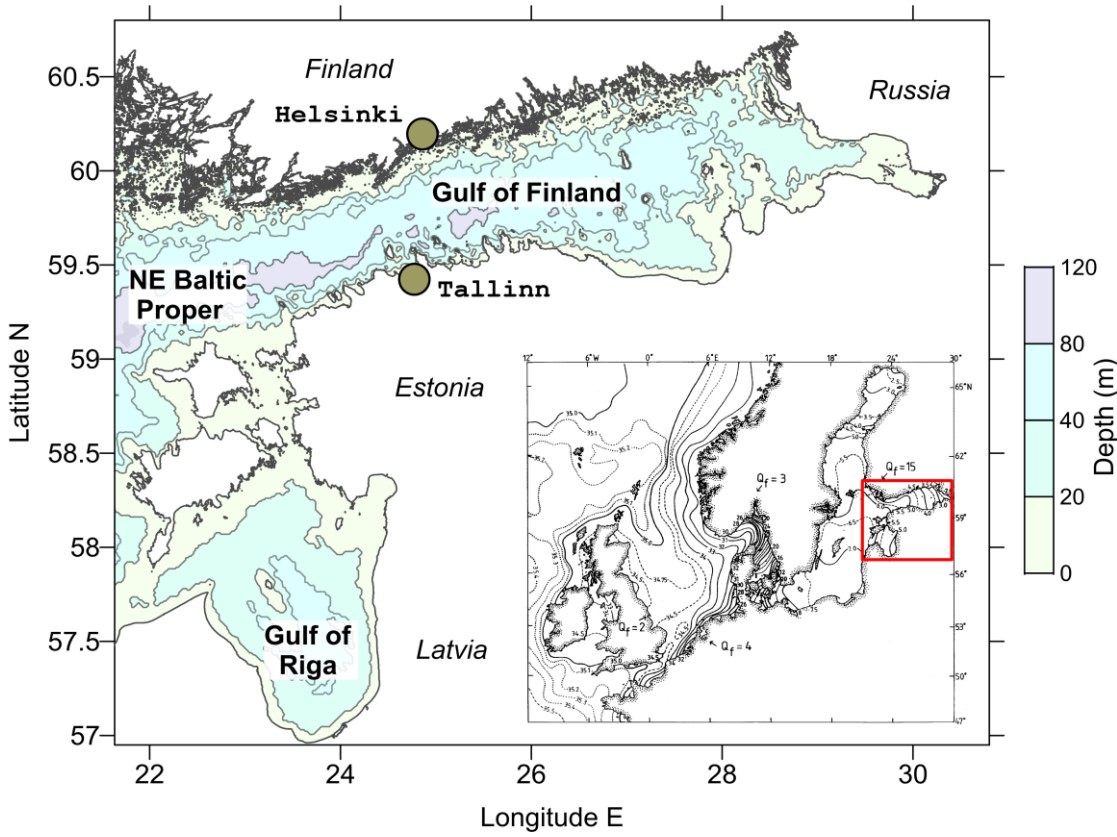

**Figure 1: Map of the study area in the NE Baltic with depth contours. Shown are the sea areas of Gulf of Finland, Gulf of Riga and part of the NE Baltic Proper.  Insert presents the map of surface salinity of the Baltic and North seas by Rohde (1998). The arrows present the mean basin-wide river discharges in 1000 m³ s⁻¹. Location of our study area is given on the insert by a red box.**

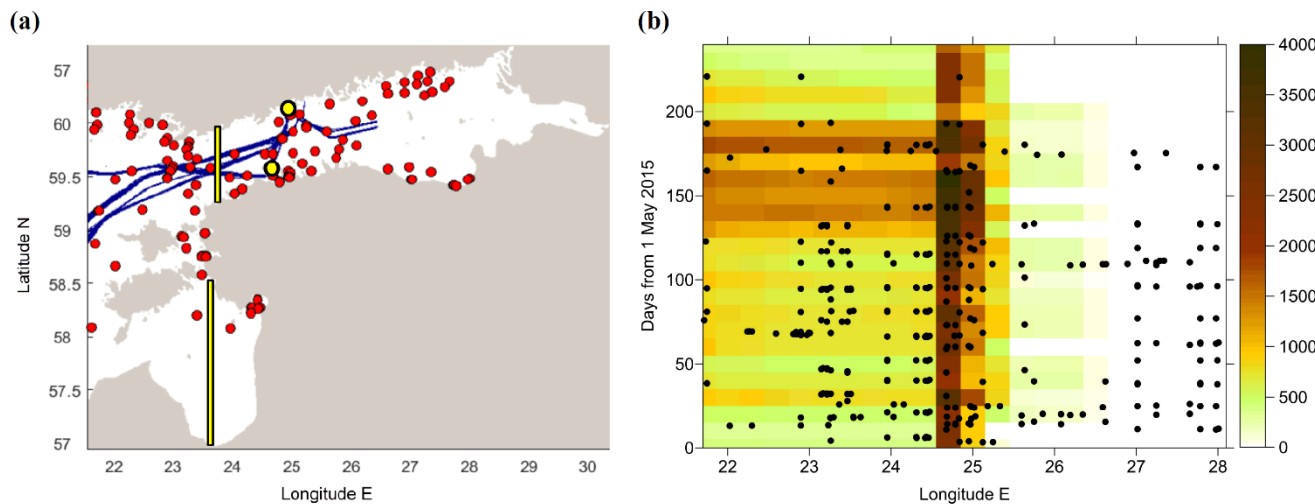

**Figure 2: Distribution of observations. (a) Map of FerryBox observation points along ship tracks (blue) and shipborne monitoring observations (red) over the study period. Shown are also the locations for time-latitude graphs and time series (black contours with yellow background). (b) Observation frequency over longitude and time. FerryBox data are shown by colour image; each image cell presents the number of initial observations over intervals of 10 days and 18' E longitude. Shipborne observations are shown by black dots.**


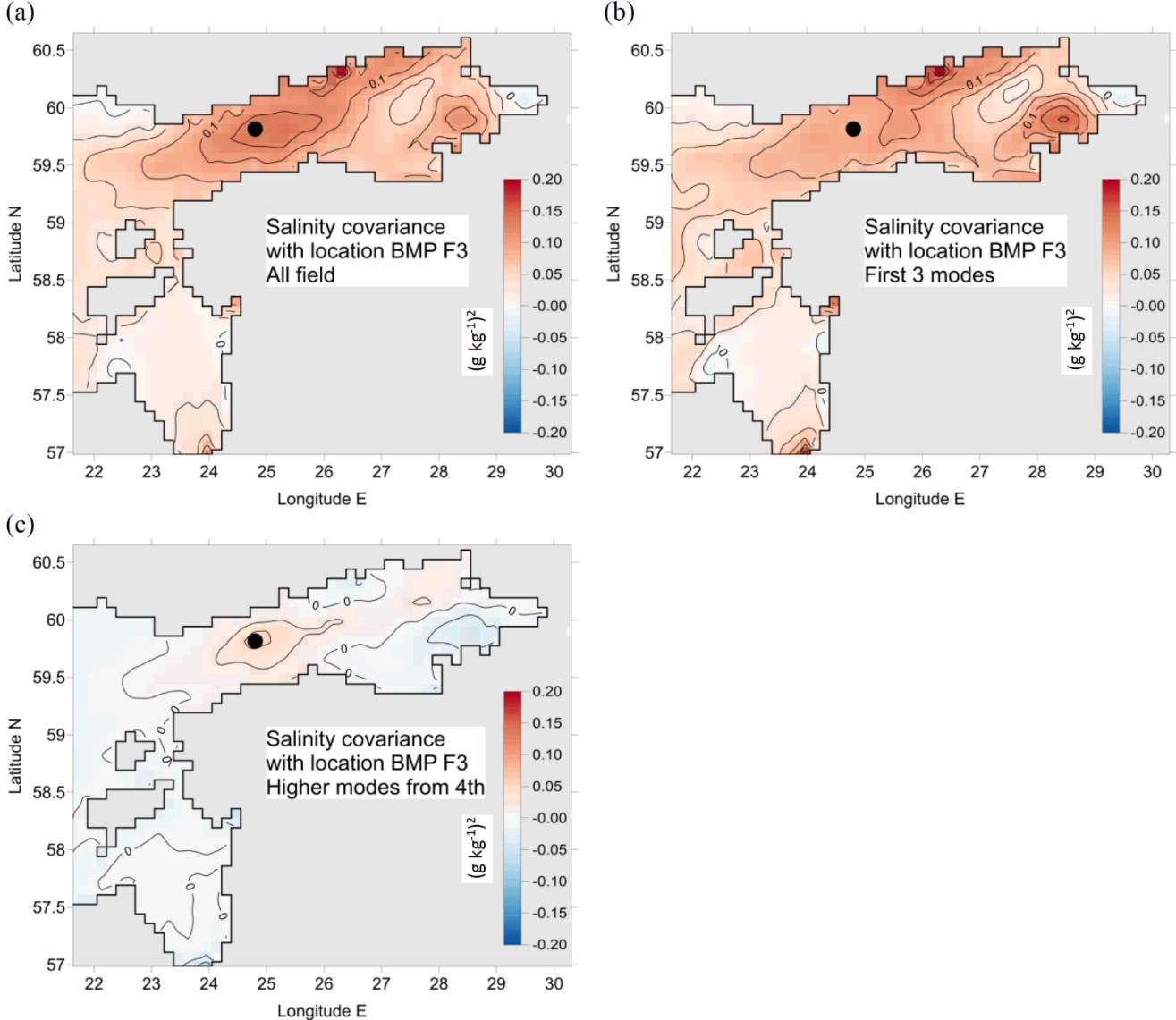

**Figure 3: Spatial covariance of SSS with the values in the grid cell near the HELCOM monitoring station BMP F3 (59.8383° N, 24.8383° E), extracted from the full covariance matrix calculated from the model data over 5 years. Covariance is decomposed by EOF modes: covariance of unfiltered data with all the modes included (a) is a sum of covariance of first three modes (b) and of the remaining higher modes, starting from the forth mode (c).**


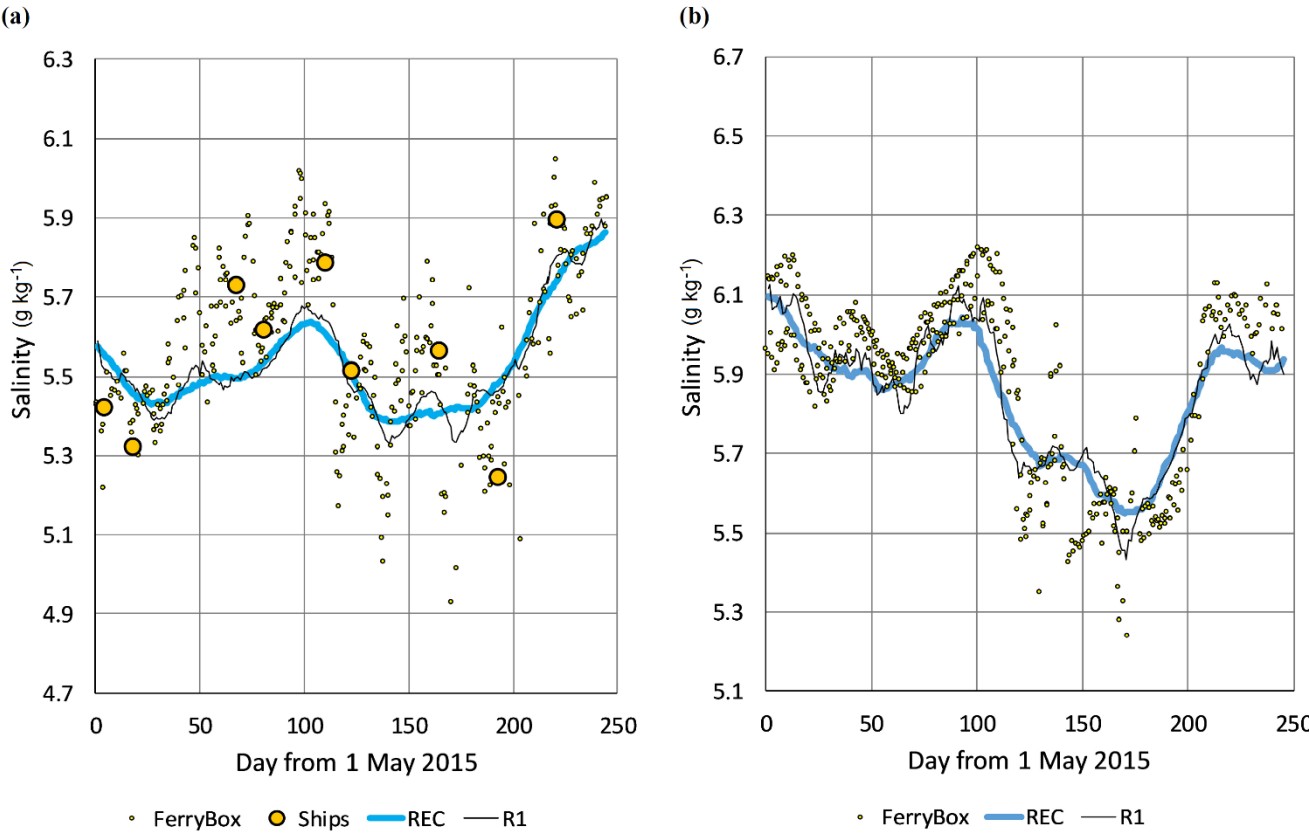

**(a)**

**(b)**


**Figure 4: Salinity time series at locations (a) 59.8383° N, 24.8383° E (HELCOM station F3) and (b) 59.794° N, 24.822° E, during the study period. Shown by dots are the observations from FerryBox and from ships (a, monitoring). Reconstructed time series, made by the time-dependent method, are given by solid lines: REC – basic option with 30 days interval, all observations in window were kept as they are; R1 – the same as previous but with time interval 10 days.**


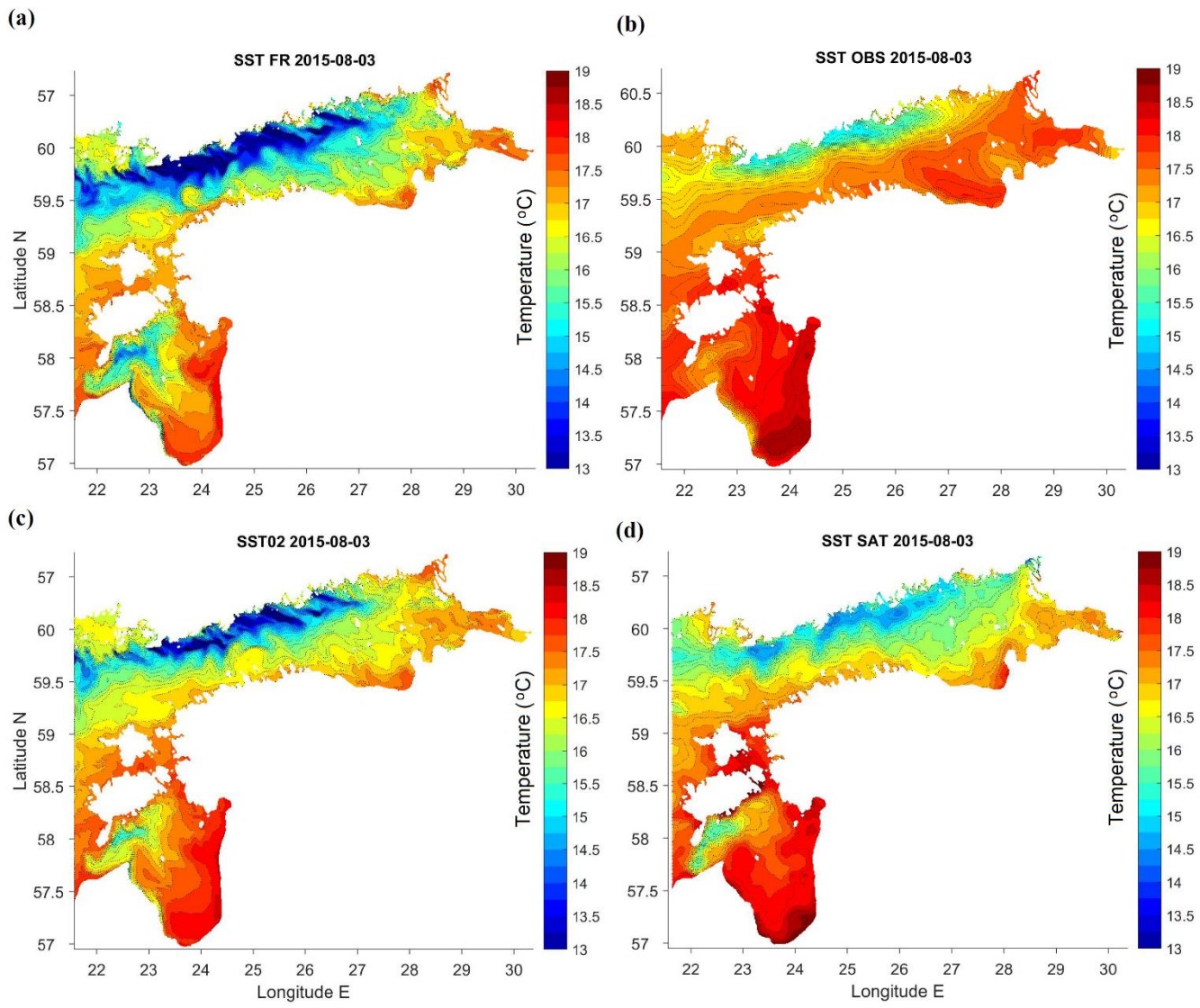

Figure 5: Maps (longitude E, latitude N) of SST in the study area on 3 August 2015: (a) free model run without DA, (b) observations reconstructed using EOF method, (c) DA with relaxation time 5 days (weight of observations 0.2), (d) satellite observations.

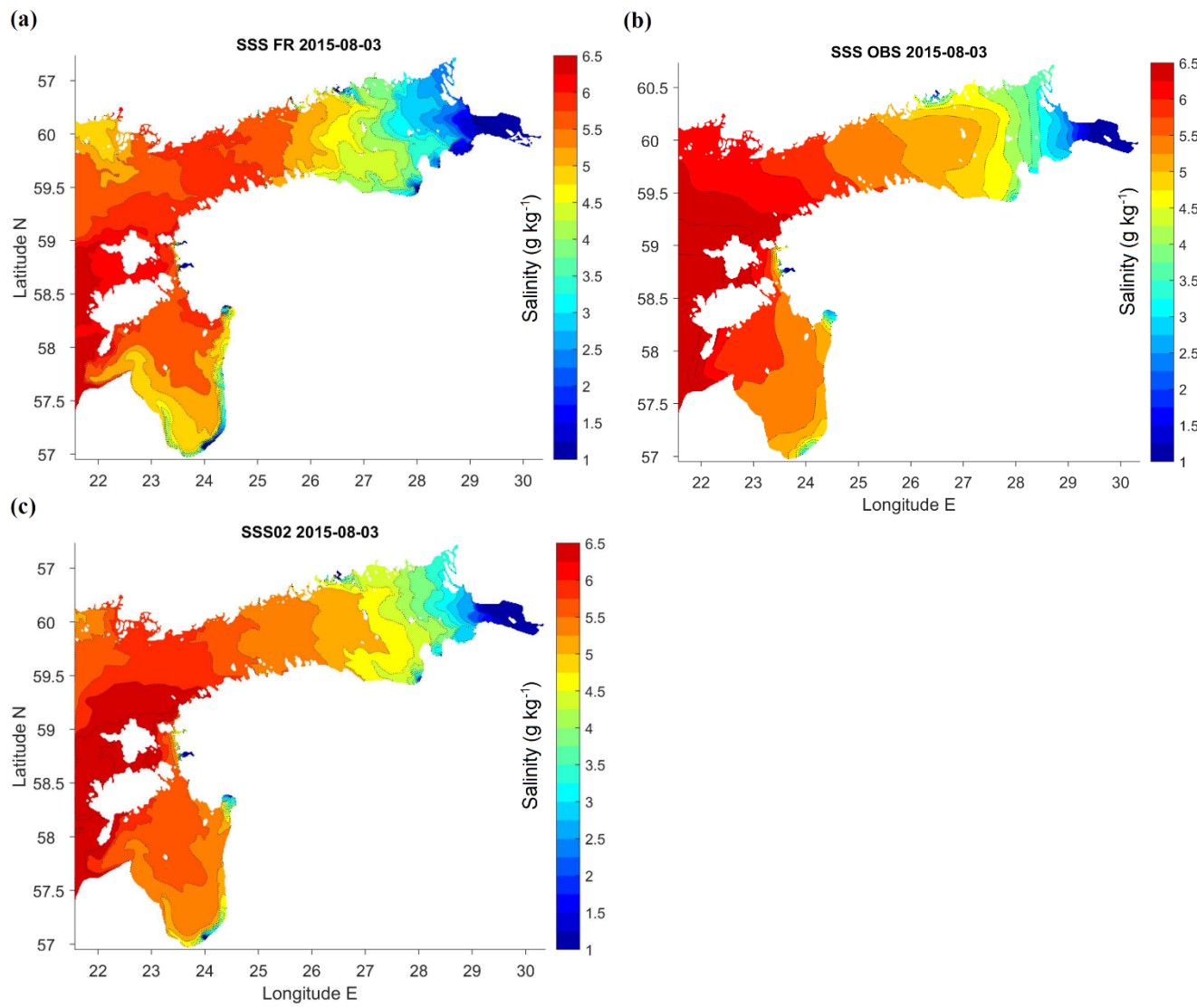


**Figure 6: Maps (longitude E, latitude N) of SSS in the study area on 3 August 2015: (a) free model run without DA, (b) observations reconstructed using EOF method, (c) DA with relaxation time 5 days (weight of observations 0.2).**

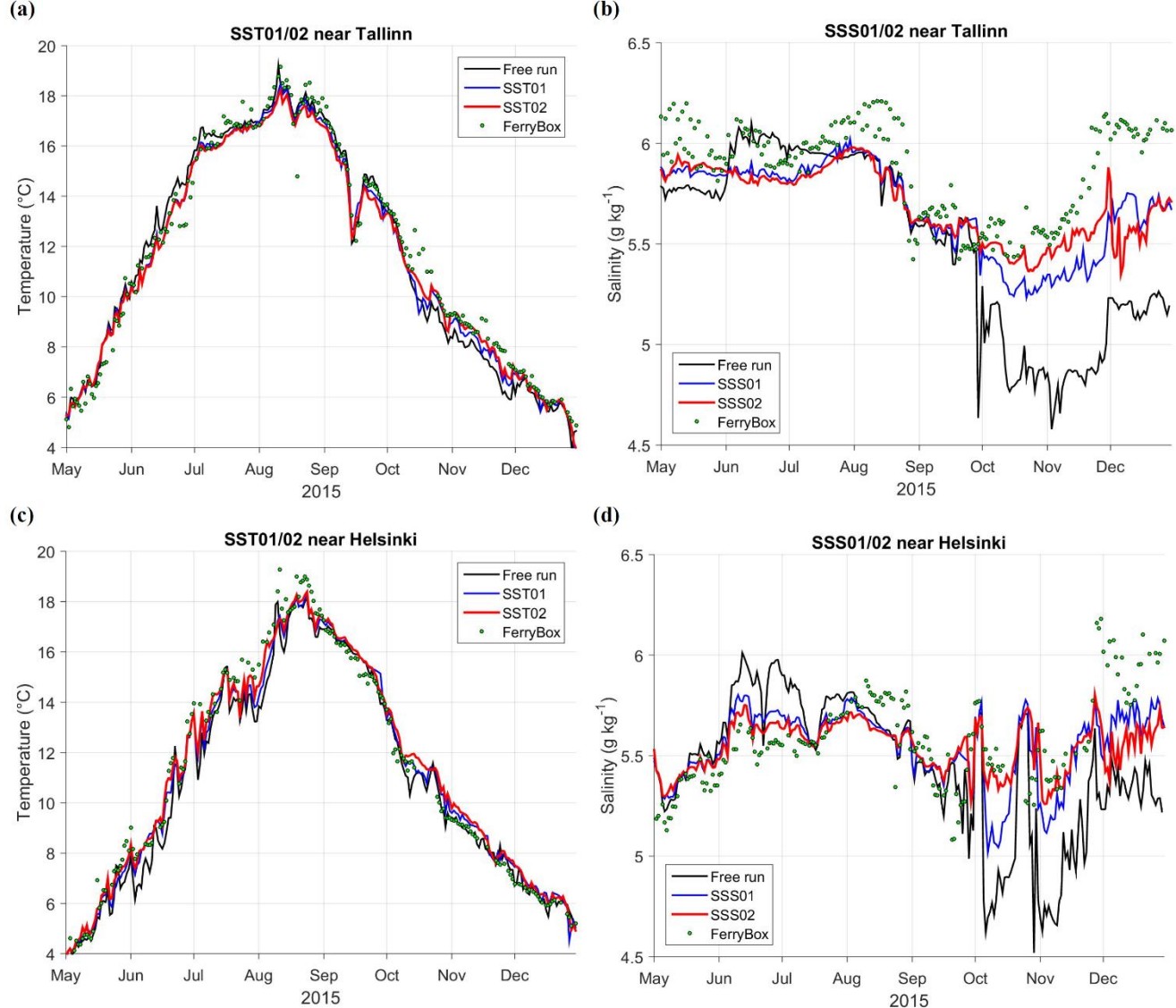

**Figure 7: Time series of SST (a, c) and SSS (b, d) near Tallinn (a, b, 59.4833° N, 24.7667° E) and Helsinki (c, d, 59.9500° N, 24.8833° E), locations shown in Fig. 2a. FerryBox data are shown by dots, black lines represent control run without DA, red lines correspond to DA with relaxation time 5 days (weight of observations 0.2), blue lines for 10 days (weight 0.1).**

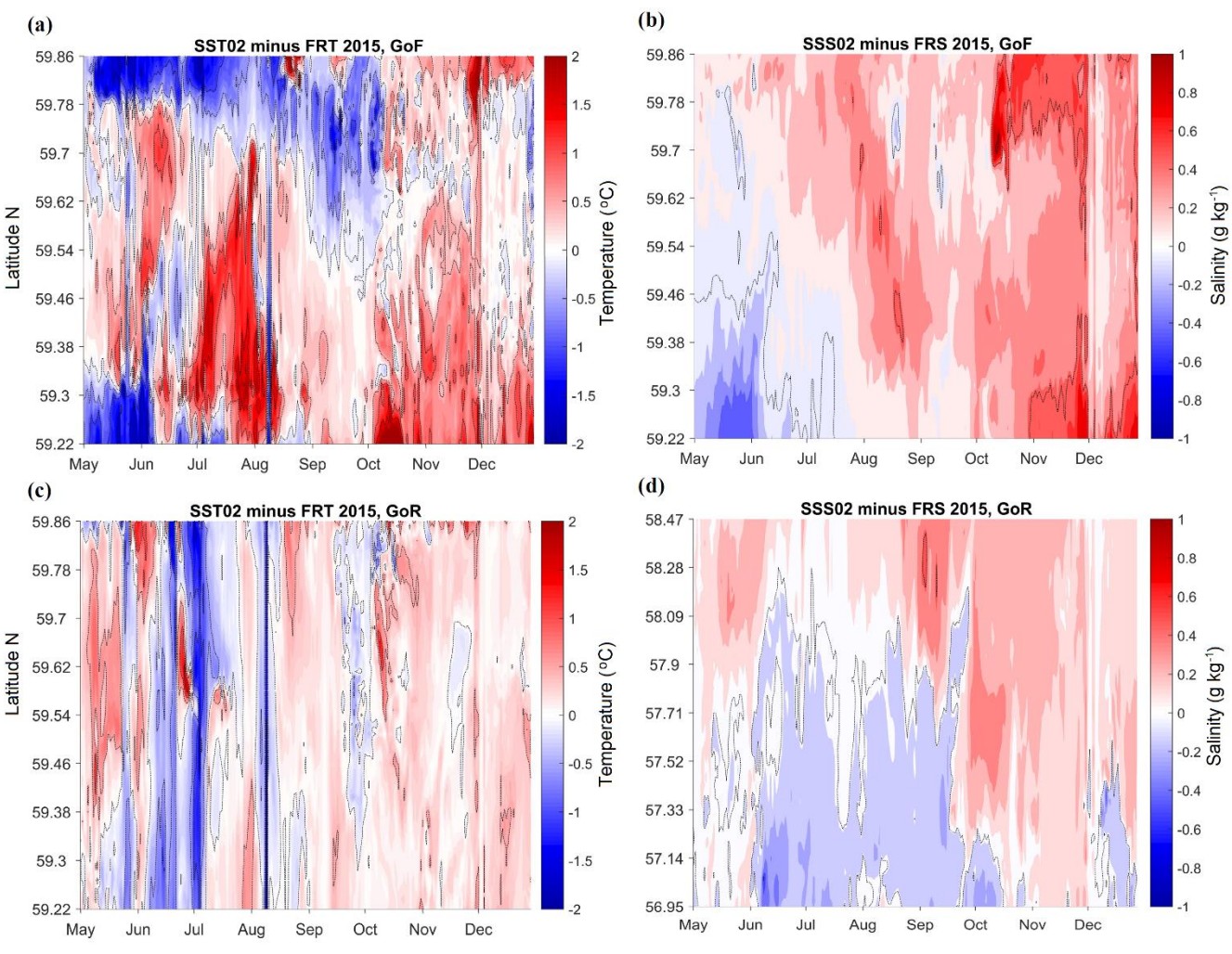


**Figure 8: Time (months of 2015) versus latitude (N) contour graph of DA anomalies of SST (a, c) and SSS (b, d) in reference to the control run (FR) without data assimilation; at longitudes 23.7166° E (a, b, Gulf of Finland) and 23.5333° E (c, d, Gulf of Riga), locations shown in Fig. 2a. DA data are given for relaxation time 5 days (weight of observations 0.2).**


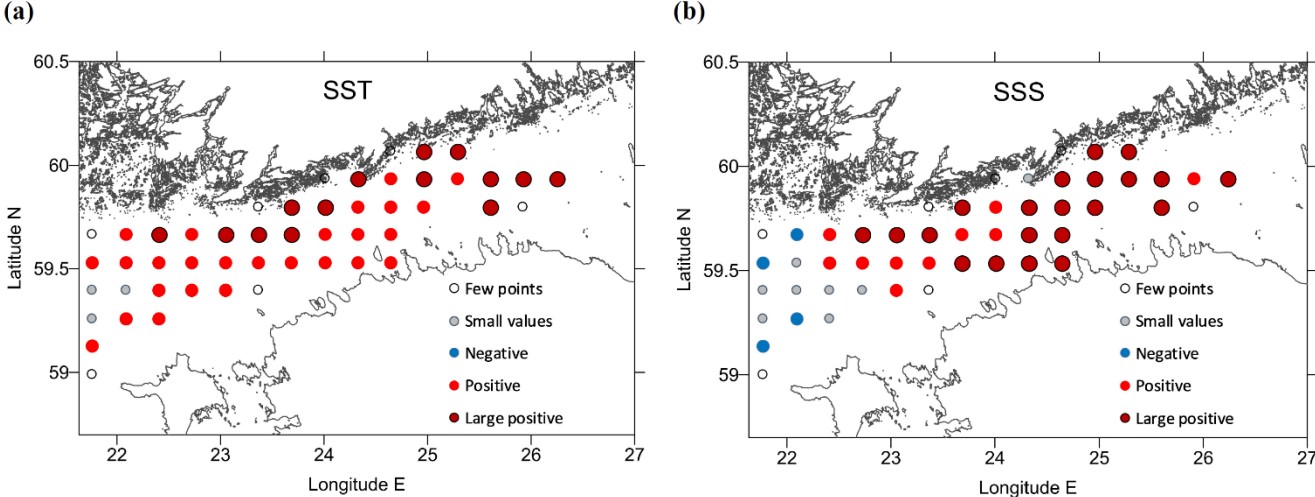

**Figure 9: Improvement of RMSD of DA compared to that of FR, both taken in reference to 110 k FerryBox observations. Comparison is made for 20 x 20 grid cells (10' N × 20' E) for SST (a) and SSS (b) over the whole study period. Legend codes: few points - less than 100 observations in a box, small values - absolute percentage change less than 10%, negative - DA RMSD growth more than 10%, positive - DA improvement (RMSD reduction) from 10% to 30%, large positive - improvement more than 30%.**
