# Peer review of "Data assimilation of sea surface temperature and salinity using basinscale EOF reconstruction: a feasibility study in the NE Baltic Sea"

_Ocean Science, 2020_

## Referee Comment (RC1) · Anonymous Referee #1 · 8 Jul 2020

The paper describes Data Assimilation experiments over a regional configuration of the NE Baltic Sea using the HBM model. The assimilated data are sparse observations of SST and SSS coming from different datasets. The analysis is a simple and classical method based on a linear regression using EOFs which are built from free simulation and no observation error are used in the analysis. A coarse grid is used to perform this analysis for physical and numerical reasons but also due to weak quantity of observation. The model is restarted using a simple nudging on SST and SSS. The results are relatively good even if the simulations with assimilation are very short in time so the stability and robustness of the results are not sure. The paper is very easy to read and the results are presented using figures of good quality. My remarks are very minor and

the paper could well fit into Ocean Science Discussion. Consequently, I would suggest a minor revision with only technical corrections.

1 Major comments (S=Section, P=Page, l=Line)

• S.2.4, l.34-35 : The authors should spatially smooth (using for instance a shapiro filter or other) the model variable psi_m before estimating (psi_m-psi_o) in order to remove "noise" in the nudging. With the present formulation, the implemented nudging tends to artificially "kill" the little scale of the model.

2 Other Comments (S=Section, P=Page, l=Line)

• S.2.3, l.185 : "that that" should be "that". The word "that" is written two times. • S.3.2.4, l.408: "Golberg" should be "Golbeck". • References : the reference for Liu should be timely ordered.

---

## Referee Comment (RC2) · Anonymous Referee #2 · 15 Jul 2020

General comments:

The manuscript describes an unusual data assimilation (DA) method which employs Empirical Orthogonal Functions (EOFs) to correct only the large-scale patterns of an ocean model for the northeastern Baltic proper. A training dataset of five years of model data was used to calculate the EOF modes. Only sea surface temperature (SST) and sea surface salinity (SSS) are considered, and the method relies on observations from a time window of up to 30 days centred around the analysis time. The authors found that the DA method is feasible to use for assimilation of SST and SSS and that it is computationally efficient.

I think the authors have made an interesting investigation of the current setup using the so-called HBM ocean model and the proposed DA technique, and I recommend the paper be published after some corrections.

General remarks:

The manuscript is well structured but not always so easy to read. I recommend a language check by a native speaker, if possible.

It is not clear whether the validation dataset was independent from the observations used in the DA process. On lines 239-240 it is stated that all observational data were used in the DA, but on lines 427 etc. it seems half of the gridded observations were reserved for validation. Please explain this more carefully. Also, even if every second gridded cell of observations were kept for validation, it is not clear to me that these observations are truly independent from the ones used in the DA, as they all originate from the same lines of FerryBox data. Is it possible to use data from certain ships for data assimilation, and use data from other ships for validation? Or is it possible to reserve the ICES data for validation, and just use the FerryBox data in the DA? Finally, the satellite-derived SST data in Figure 4 (d) is used to discuss the results in a qualitative way only; why not use it (together with similar satellite data) also for a statistical validation? In short, I would like to see a more careful validation using truly independent observations.

The DA method relies on the use of EOF reconstructions of SST and SST. Please show some examples of the reconstructions that were used in the DA.

It has been shown that the DA method works in a "reanalysis mode", in which observations "from the future" can be used in a time window up to 30 days wide, centred on the analysis date. I can see a problem when this method is used in forecast mode, where observations mainly from the past several days can be used. Is it enough to have a time window of, say, six hours? Please discuss this more.

Detailed, technical comments:

(I omit page numbers as the line numbers are unique)

l.8-9: "...based on covariance estimates from long..." (too many "the")

l.10; "...on a regular grid."

l.35: "...do not presently include DA..." (word order)

l.66: "Baltic proper" (should not be spelled with capital P; see also other occurrences in the manuscript)

l.130: "Two sets of compressed (averaged) FerryBox..." (for clarity); also on line 133.

l.134: "...it was chosen not to..." (word order)

l.142: "...data was too irregular..."

l. 146: "...time fixed..." How do you mean? Are they time-independent? Or just interpolated to pre-determined, fixed times, e.g. 00 UTC?

l.170: "observation operator Hi..."

l.179-181: "In practice, ..." I don't quite understand this sentence; please rewrite...

l.183: "...are not made at the same time." (simpler)

l.183: "...to cover a larger sea area..."

l.184: "...observation operator Hp..."

l.185: remove one instant of "that"

l.204: What happens if there is only one observation available?

l.215: "...without DA..." This puzzles me, as the analysis field depends on DA..?

l.224: "...can easily be included..." (word order)

l.271: "...frequently became unrealistic."

l.274: "The time windows for experiments (a) and (b) were selected to be 10..."

l.317: "...revealed a deep... ...and a shallower..."

l.344: "...strips of lower salinity..."

l.395: "...when DA had decreased the FR temperature..." FR = free run; so DA cannot affect the temperature... I think you mean "decreased the temperature in the SST01 and SST02 datasets"...?

l.411: Is it the centred RMSD that is being used? Why not the usual RMSD? The centred RMSD is calculated after removing the bias; which do you mean?

l.418: "...so many observations."

---

## Referee Comment (RC3) · Anonymous Referee #3 · 16 Jul 2020

Review of the manuscript

"Data assimilation of sea surface temperature and salinity using basin scale EOF reconstruction: a feasibility study in the NE Baltic Sea"

Authored by

M. Zujev, J. Elken and P. Lagemaa

**General Comments**

The paper addresses an important issue, which is the estimation of salinity and temperature for the Baltic Sea using a combination of model data and observations.

The method is based on a two-step approach, in which sparse observations are interpolated using an EOF technique and subsequently a relaxation method is applied for the assimilation into the model.

The method seems to have some potential for the assimilation of FerryBox data, where interpolation to 2D grids make sense, if longer time scales are considered.

There are a couple of concerns, which should be addressed:

- The method assumes in the interpolation step, that the covariance structure of the model is correct. This should be discussed more – in particular the limitations caused by this
- The previous point is related to a discussion of the main model error sources, which is missing as well
- Observation errors are not discussed at all – this needs to be justified and discussed
- We missed some discussion on the potential of the method to improve forecasts
- We would furthermore appreciate some discussion about the implications of the assimilation on the model dynamics (e.g., vertical density structure, in particular stability)
- It is not really clear, why the authors did not apply a more standard technique, with a more solid theoretical basis. A straightforward approach would be to use a low rank model error covariance matrix based on the presented EOF decomposition in the standard Kalman analysis equation. This would then also include observation errors and avoid the two steps required in the presented technique.

The presentation of the material should be improved. There are deficiencies, in particular with regard to putting the study into context of existing methods and motivating the selected approach. If "computational effort" is the main point, then this has to be quantified better.

There are quite a view grammar problems and a native speaker should proofread the text.

We recommend publication after major revisions.

**Specific Comments**

Abstract

We think it would be better to structure the abstract such, that more general information (what is done ?) comes first and specific results follow after that

Please explain acronym RMSD

I think it is more common to say "dominating EOF modes" instead of "gravest", but that should be checked by a native speaker

Introduction

Page 1, line 22: please reformulate "discrepancies of"

Page 2, line 34: replace "then" by "the"

Data and methods

Page 3, line 72 : "whichever" instead of "which"

Page 3, line 79: "… from the halocline …" please reformulate

Page 3, line 87: "better grid cells" instead of "points"

It would be good to learn more about the vertical discretization of the model, e.g., how thick is the surface layer.

Page 4, line 95, maybe better "grid resolution" instead of "grid step"

Fig. 2a: Please change color of FerryBox tracks – it cannot be distinguished from land.

Please add information on the water depth the FerryBox observations are usually taken.

Page 5: It was not clear, how you interpolate the FerryBox data to a 2D grid. Please explain in more detail.

Page 5, line 142 : Did you mean "… too irregular …" ?

Page 5, lines 146-150 : this paragraph is hardly understandable – please reformulate.

Page 6, line 160: "the" instead of "then"

Section 2.3

The entire section is unfortunately quite messy and confusing, although (as far as I understand) the method is quite basic. The authors have to explain all symbols and indices with much more care. Also, what is a vector or matrix (what size?) and what is a scalar?

Page 6, line 185 : "that that"

The basic assumption, if you use EOFs for interpolations like you do, is that the covariance structure of the model is correct – this should be stated more explicitly and discussed a little.

You could have included observation errors in this interpolation exercise. I guess your assumption at the moment is, that the observations are 100% correct ? - please comment

Eq. 1: you assume that this matrix actually has an inverse – please comment.  If the matrix is close to singular, you run into numerical problems as well.

Section 2.4

I guess eq. 1 is a continuous equation, which in its original form should be solved using the internal model time. I assume that you get eq. 4, if you replace the model time step by the assimilation time step – please explain more

I had problems to figure out how big the assimilation time step in the experiments actually is – please use consistent notation for cititical parameters (e.g. time steps) throughout the document.

Page 8: line 223 : "The DA method … is analogical …"    I don't think this is true in this generality, because it seems you don't consider observation errors at all. – please comment. The resemblance with 4DVAR is remote, because there is no model dynamics included in the minimization of the cost function.

Page 8, line 239:  "… artificial split …"   I don't understand this sentence, because this "split" is a standard approach to validate assimilation techniques.

Page 7, line 217: " … since extensive use has been made …".  This is I guess the critical point. The classical approach in an assimilation filter is to combine observations and the model state using covariance information on model errors at each analysis time step. In your approach there are no covariances of model errors. Instead, you use covariances of the background statistics for the interpolation. If you used a scaled version of the background covariance as a proxy for the model error covariance in a classical filter approach, you would probably end up with similar results, but with a more solid theoretical foundation. Anyway, as pointed out in the general comments, the method has to be put into the context of existing methods in a better way.

Section 3.1

Page 9, line 275: This is interesting; why don't you show the EOFS computed in your study ?

Page 13, line 408: The skill is often defined in relation to a reference run (e.g. the free run). In the case of the standard forecast skill, it is a dimensionless number – please check.

Page 16: "There are obvious extensions  … layers …"
This is, where it gets interesting, because the vertical structure of different model variables (temperature, salinity, etc) is a particular challenge and your assumption about the correctness of model covariances may become a problem (e.g., if the mixed layer thickness in the model is not correct)

Figure 8: It would be interesting to see the absolute differences between observations and the assimilation run and the same for the free run (these differences should reflect both observation and model errors).

---

## Author Response (AR1)

We thank the referees who have made excellent work in going through the details of our submitted MS and made very constructive remarks and corrections. Our detailed step-by-step responses to each of the Referee #1 comments or questions are given below.

We have revised the MS, with the following main points.

- The main points of the EOF reconstruction and the found modes were presented too briefly, relying mainly on the reference Elken et al. (2019). In the revised MS, additional important issues have been included in the compact form (hopefully not repeating the already published MS).
- Justification for the large-scale EOF DA method, in comparison with other well-known DA methods, has been refined.
- Data transformations between the fine and coarse grids have been more carefully presented.
- Unfortunately, the issue of observational errors has not been included in the initial MS. It is now included in the revised MS.
- Presentation of DA validation has been reformulated and discussed in more details.
- Possibilities of the method regarding operational forecast (with assimilating only the past data) have been discussed.

Suggested technical corrections have been included as well.

**Anonymous Referee #1**

**Comments and questions in bold**
Response by the authors in normal
Line and Figure numbers taken from first submission

**The paper describes Data Assimilation experiments over a regional configuration of the NE Baltic Sea using the HBM model. The assimilated data are sparse observations of SST and SSS coming from different datasets. The analysis is a simple and classical method based on a linear regression using EOFs which are built from free simulation and no observation error are used in the analysis. A coarse grid is used to perform this analysis for physical and numerical reasons but also due to weak quantity of observation. The model is restarted using a simple nudging on SST and SSS. The results are relatively good even if the simulations with assimilation are very short in time so the stability and robustness of the results are not sure. The paper is very easy to read and the results are presented using figures of good quality. My remarks are very minor and the paper could well fit into Ocean Science Discussion. Consequently, I would suggest a minor revision with only technical corrections.**

**1 Major comments (S=Section, P=Page, l=Line)**
**S.2.4, l.34-35: The authors should spatially smooth (using for instance a shapiro filter or other) the model variable psi_m before estimating (psi_m-psi_o) in order to remove "noise" in the nudging. With the present formulation, the implemented nudging tends to artificially "kill" the little scale of the model.**

Reconstruction psi_o is made on the coarse grid and transfer to the fine grid is smooth, using bilinear interpolation. Adding a smooth field to the fine-scale model results indeed damps the small-scale motions. In the first experiments, we used 10x10 grid points average filter (not the Shapiro filter) to find the coarse grid values from fine grid results, and applied a bilinear filter to find the deviations from the coarse grid. Those deviations were frozen during the given DA time step. After modifying

the coarse grid fields using observational reconstruction, these deviations were added to the result in order to obtain a fine grid analysis field.

In our study area, meso- and small-scale features are in a continuous generation and damping balance, therefore damping by relaxation to the smooth observation fields does not smooth out the fine grid variability, as can be seen from Figs. 4 and 5. Therefore, we used the simplest approach in the feasibility study.

We have added explanations on the damping problem into the revised MS. The paragraph on lines 220-222 has been replaced to:

"In practical calculations, SST and SSS observational data were reconstructed on the coarser 5' N $\times$ 10' E grid and interpolated/extrapolated by bilinear procedure to the finer 0.5' N $\times$ 1' E model grid. Such simple transition of data from coarse to finer grid includes smoothing, since $\psi^o$ lacks the details that are present on the finer grid. We have tested that the effect of added smoothing is smaller than the physical diffusion. In our study area, generation of meso- and small-scale features is of high intensity; therefore relaxation to the smooth observation fields does not apparently damp the fine grid variability. The approach of using two grids with different resolutions is justified by irregular distribution of observations; reliable estimation is possible only for large-scale patterns of SST and SSS fields; the computationally more efficient coarser grid resolves these patterns with enough details."

**2 Other Comments (S=Section, P=Page, l=Line)**
**S.2.3, l.185: "that that" should be "that". The word "that" is written two times.**

Corrected.

**S.3.2.4, l.408: "Golberg" should be "Golbeck".**

Corrected.

**References: the reference for Liu should be timely ordered.**

Unfortunately, it seems that this remark cannot be accepted since the manuscript preparation guidelines https://www.ocean-science.net/for_authors/manuscript_preparation.html say: If there is more than one work by the same first author, their papers are listed in the following order: (1) single author papers (first author), followed by (2) co-author papers (first author and second author), and finally (3) team papers (first author et al.).

We thank the referees who have made excellent work in going through the details of our submitted MS and made very constructive remarks and corrections. Our detailed step-by-step responses to each of the Referee #2 comments or questions are given below.

We have revised the MS, with the following main points.

- The main points of the EOF reconstruction and the found modes were presented too briefly, relying mainly on the reference Elken et al. (2019). In the revised MS, additional important issues have been included in the compact form (hopefully not repeating the already published MS).
- Justification for the large-scale EOF DA method, in comparison with other well-known DA methods, has been refined.
- Data transformations between the fine and coarse grids have been more carefully presented.
- Unfortunately, the issue of observational errors has not been included in the initial MS. It is now included in the revised MS.
- Presentation of DA validation has been reformulated and discussed in more details.
- Possibilities of the method regarding operational forecast (with assimilating only the past data) have been discussed.

Suggested technical corrections have been included as well.

**Anonymous Referee #2**

**Comments and questions in bold**
Response by the authors in normal
Line and Figure numbers taken from first submission

**General comments:**
**The manuscript describes an unusual data assimilation (DA) method which employs Empirical Orthogonal Functions (EOFs) to correct only the large-scale patterns of an ocean model for the northeastern Baltic proper. A training dataset of five years of model data was used to calculate the EOF modes. Only sea surface temperature (SST) and sea surface salinity (SSS) are considered, and the method relies on observations from a time window of up to 30 days centred around the analysis time. The authors found that the DA method is feasible to use for assimilation of SST and SSS and that it is computationally efficient.**

**I think the authors have made an interesting investigation of the current setup using the so-called HBM ocean model and the proposed DA technique, and I recommend the paper be published after some corrections.**

**General remarks:**
**The manuscript is well structured but not always so easy to read. I recommend a language check by a native speaker, if possible.**

We plan additional language check.

**It is not clear whether the validation dataset was independent from the observations used in the DA process. On lines 239-240 it is stated that all observational data were used in the DA, but on lines 427 etc. it seems half of the gridded observations were reserved for validation. Please explain this more carefully.**

The sentence on lines 239-240 was deleted as not necessary in this location.

In our method, DA depends only on the accuracy of observational gridded maps that were pre-calculated prior to the DA experiments. All the observations were included in the calculation. Experiments were made with options for reconstruction. Reducing the number of observation "boxes" by a factor of two gave nearly the same reconstruction results as the reconstruction with a full set of observational data. Pointwise comparison of SSS reconstruction over the full study period is presented in the Fig. X1 inserted below.

[Figure]

**Figure X1.** Scatter plot of all reconstructed SSS grid values over the study period: reconstruction with all observations included versus reconstruction with every coarse grid average omitted. Shown are the characteristics of linear regression.

The figure is not included in the MS, but the numerical estimates are given. The sentences on lines 426-431 are modified and replaced to:

"The experiments which took every second available observation "box" into account (this resulted in mean sampling interval along ship tracks about 20 km instead of 10 km) revealed that performing DA during the study period with reduced data set (6.5 k averaged observation data instead of 13 k) changed RMSD of SST by only 1% and of SSS by 2%, whereas the RMSD values were 0.05 °C for SST and  0.027 g kg$^{-1}$ for SSS. It was evaluated over the full time span and domain using 182 k coarse grid cells; correlation between the data sets was higher than 0.999. We have also checked reconstruction results with FerryBox data only, excluding the data from shipborne monitoring stations. Compared with the full data set, largest (but still minor) differences with RMSD of SSS up to 0.03 g kg$^{-1}$ were found in the Gulf of Riga and the eastern Gulf of Finland, where FB data were missing."

**Also, even if every second gridded cell of observations were kept for validation, it is not clear to me that these observations are truly independent from the ones used in the DA, as they all originate from the same lines of FerryBox data. Is it possible to use data from certain ships for data assimilation, and use data from other ships for validation? Or is it possible to reserve the ICES data for validation, and just use the FerryBox data in the DA? Finally, the satellite-derived SST data in Figure 4 (d) is used to discuss the results in a qualitative way only; why not use it (together with similar satellite data) also for a statistical validation? In short, I would like to see a more careful validation using truly independent observations.**

If the new observation points are separated by a distance of positive significant correlation, then the observational results are not truly independent. It is principally possible to use data from certain ships for data assimilation, and use data from other ships for validation, but the problem is that different ships cover different areas with different time intervals. For example, excluding the data from FinnMaid (Helsinki – Travemünde, Table 1) means that data from the Baltic Proper south from the Helsinki – Stockholm and Tallinn – Stockholm lines will be missing. It can be expected that different combinations of exclusion will give different results due to different geographical coverage. Sorting out such variations would require a large number of new time-consuming calculations, which is not reasonable for the first feasibility study of the method. We are updating both the computing facilities and the core operational forecast model, and plan longer DA experiments with more validation options in the near future.

There were about 370 shipborne SST and SSS observations available, originating from about 80 spatially separated stations. This is a very small amount compared to the FerryBox data and therefore the shipborne statistics is not well comparable to that of the whole data set. We have done SSS reconstruction experiments as shown in Fig. X2. It was found that shipborne ICES data had only a minor effect on the results, since the large-scale variability with high spatial correlation dominates in the region.

We discussed earlier between ourselves about the possibility of comparison of SST DA with remote sensing results. We came to the opinion that this would bring too many details to our feasibility study, since it would also include non-trivial aspects of comparison of FerryBox data with different remote sensing products. This comparison can be done in a later stage

All the suggestions proposed are very valuable and we plan to perform such thorough validation studies at a later stage, when this DA system is going to be implemented in everyday forecast procedures.

We have added following text to the MS on line 431 before the last sentence of the paragraph:

 "We have also checked reconstruction results with FerryBox data only, excluding the data from shipborne monitoring stations. Compared with the full data set, largest (but still minor) differences with RMSD of SSS up to 0.03 g kg$^{-1}$ were found in the Gulf of Riga and the eastern Gulf of Finland, where FB data were missing."

[Figure]

[Figure]

**Figure X2.** Scatter plots of reconstructed SSS time series at six locations shown on the map (on top of the panel). Shown are reconstruction based on all observational data versus reconstruction based on FerryBox data only.

**The DA method relies on the use of EOF reconstructions of SST and SST. Please show some examples of the reconstructions that were used in the DA.**

Examples of the reconstructions were added for 3 August 2015, to be compared with the maps in Figs 4 and 5 (former numbering).

**It has been shown that the DA method works in a "reanalysis mode", in which observations "from the future" can be used in a time window up to 30 days wide, centred on the analysis date. I can see a problem when this method is used in forecast mode, where observations mainly from the past several days can be used. Is it enough to have a time window of, say, six hours? Please discuss this more.**

Value of the time window depends on the spatio-temporal characteristics of the studied field and on the observational network. It is necessary that there are critical number of observations (say, 6 observations) available, in order to find observational EOF amplitudes from dominating modes. Remind that therefore we can detect only the large-scale patterns. With SST and SSS data, the amplitudes of dominating modes have temporal correlation scale generally more than 60 days, except for the SST "upwelling mode" which has about 15 days. We selected a centred time window of 30 days, although 10 days worked also well in most of the dates (some dates were dropped out because of too little data). If there are hourly time series available (like in recent years, there is data from buoy stations and gliders), it is possible to reduce the time window significantly, why not to try 6 hours. We did not consider in this study the sea level, but there is good hourly data available over all the coasts of the Baltic.

Time-dependent EOF reconstruction method enables the option to use only the past data as during the operational forecast mode. Time sequence of past observations is used to determine the rate of change of amplitudes, assuming that within the time window the amplitudes depend linearly on

time. There are good examples shown by Elken et al. (2019). However, extensive tests for using the past data only are not in the scope of the present feasibility study.

We have added following paragraph after line 487:
"We have tested the EOF-based DA in centred time window of 30 days, based mainly on available FerryBox data during the study period. As shown by reconstruction experiments by Elken et al. (2019), the time-dependent method can also work with backward observations as if it occurs during operational forecasts. When more observations become available, for example from new automated buoy stations, Argo floats and gliders, the time window can be shortened. Full covariance matrix estimated from the model results is the backbone of the EOF DA method. Prior and/or complementary to implementation of the method into operational practice, detailed covariance studies using results from multiple models could be useful, as well as additional reconstruction and DA studies using more data sources over longer periods."

**Detailed, technical comments:**
**(I omit page numbers as the line numbers are unique)**
**l.8-9: "...based on covariance estimates from long..." (too many "the")**

Corrected.

**l.10; "...on a regular grid."**

Corrected.

**l.35: "...do not presently include DA..." (word order)**

Corrected.

**l.66: "Baltic proper" (should not be spelled with capital P; see also other occurrences in the manuscript)**

Both versions, "Baltic Proper" and "Baltic proper" are used in the scientific literature. Our historical preference of using "Baltic Proper" is partly reasoned by the HELCOM nomenclature of the sub-regions of the Baltic Sea, see https://helcom.fi/wp-content/uploads/2019/06/Implementation-of-the-BSAP-2018.pdf. We keep the term as it was written, "Baltic Proper".

**l.130: "Two sets of compressed (averaged) FerryBox..." (for clarity); also on line 133.**

Corrected.

**l.134: "...it was chosen not to..." (word order)**

Corrected.

**l.142: "...data was too irregular..."**

Corrected.

**l. 146: "...time fixed..." How do you mean? Are they time-independent? Or just interpolated to pre-determined, fixed times, e.g. 00 UTC?**

The time-fixed approach uses EOF amplitudes that do not depend on time. Later, time-dependent amplitudes consider EOF amplitudes and their time derivatives within a selected time interval. For clarity, the sentence has been reformulated to:

"The basic option of EOF reconstruction uses at each DA time step time-fixed amplitudes, encountering the observations spanning over certain time (which can be longer than DA time step) that are transferred to the fixed times by some interpolation or filtering/averaging procedure."

**l.170: "observation operator Hi..."**

Corrected as suggested. Although, in most cases the operator takes the form of a matrix.

**l.179-181: "In practice, ..." I don't quite understand this sentence; please rewrite...**

This sentence has been deleted. The earlier sentence has been modified:

"Experiments with pseudo-observations (Elken et al., 2019) revealed that the values of $\hat{\mathbf{a}}_i$ of dominating $L$ modes should match the limits derived from statistics of $\tilde{\mathbf{a}}_i$, whereas higher modes with outlying amplitudes should be neglected."

**l.183: "...are not made at the same time." (simpler)**

Corrected.

**l.183: "...to cover a larger sea area..."**

Corrected.

**l.184: "...observation operator Hp..."**

Corrected.

**l.185: remove one instant of "that"**

Corrected.

**l.204: What happens if there is only one observation available?**

With one observation available only, the amplitude of only the 1st EOF mode can be estimated, but most probably it will not fit to the statistical limits and have to be neglected. We have excluded the times when the number of observations was less than six (line 272).

**l.215: "...without DA..." This puzzles me, as the analysis field depends on DA..?**

The phrase has been rewritten:

"... calculated from the previous analysis field $\psi^{a-1}$  using only the model operator $F$ without DA during this time step,"

**l.224: "...can easily be included..." (word order)**

Corrected.

**l.271: "...frequently became unrealistic."**

Corrected.

**l.274: "The time windows for experiments (a) and (b) were selected to be 10..."**

Corrected.

**l.317: "...revealed a deep... ...and a shallower..."**

Corrected.

**l.344: "...strips of lower salinity..."**

Corrected.

**l.395: "...when DA had decreased the FR temperature..." FR = free run; so DA cannot affect the temperature... I think you mean "decreased the temperature in the SST01 and SST02 datasets"...?**

Partially incorrect sentence was rewritten. The new sentence is:

"In the Gulf of Riga, SST increase dominated throughout the study period, but it was interrupted occasionally by basin-wide events when DA had decreased the  temperature compared to the results from FR."

**l.411: Is it the centred RMSD that is being used? Why not the usual RMSD? The centred RMSD is calculated after removing the bias; which do you mean?**

We have used centred root-mean-square difference when comparing observations and model results. RMSD, giving also explanation "standard deviation of differences at all the observation points is denoted as centred RMSD". It means that the average difference between observations and model (bias) is not included in the centred RMSD. Many recent studies analyse the model results using Taylor (2001) diagram, which is based on the centred RMSD dependence on variances and correlation; our choice was made to have compatibility with such studies that consider bias and RMSD separately. Explanations of the RMSD acronym were checked throughout the MS and unified. In particular, the acronym RMSD was omitted when describing the use of least squares method.

**l.418: "...so many observations."**

The sentence has been reformulated:

"Areas with lower salinity in the eastern Gulf of Finland and in the Gulf of Riga  had only a small number of observations."

We thank the referees who have made excellent work in going through the details of our submitted MS and made very constructive remarks and corrections. Our detailed step-by-step responses to each of the Referee #3 comments or questions are given below.

We have revised the MS, with the following main points.

- The main points of the EOF reconstruction and the found modes were presented too briefly, relying mainly on the reference Elken et al. (2019). In the revised MS, additional important issues have been included in the compact form (hopefully not repeating the already published MS).
- Justification for the large-scale EOF DA method, in comparison with other well-known DA methods, has been refined.
- Data transformations between the fine and coarse grids have been more carefully presented.
- Unfortunately, the issue of observational errors has not been included in the initial MS. It is now included in the revised MS.
- Presentation of DA validation has been reformulated and discussed in more details.
- Possibilities of the method regarding operational forecast (with assimilating only the past data) have been discussed.

Suggested technical corrections have been included as well.

**Anonymous Referee #3**

**Comments and questions in bold**
Response by the authors in normal
Line and Figure numbers taken from first submission

**General Comments**
**The paper addresses an important issue, which is the estimation of salinity and temperature for the Baltic Sea using a combination of model data and observations.**

**The method is based on a two-step approach, in which sparse observations are interpolated using an EOF technique and subsequently a relaxation method is applied for the assimilation into the model.**

**The method seems to have some potential for the assimilation of FerryBox data, where interpolation to 2D grids make sense, if longer time scales are considered.**

**There are a couple of concerns, which should be addressed:**
- **The method assumes in the interpolation step, that the covariance structure of the model is correct. This should be discussed more – in particular the limitations caused by this**

Problems of EOF reconstruction have been considered by Elken et al. (2019). We cite: SST and SSS results are rather well validated by observations and the model-based covariance patterns can be considered trustful. /// Fu et al. (2011) compared covariance patterns from modeled SST and satellite SST, and found them agreeing well. CMEMS QUID report has presented validation of SSS against FerryBox data, showing that the SSS patterns were well simulated by the model. In deeper layers, however, there is usually a larger spread between different model results.

Main differences between actual and model-based covariance estimates are expected within very short term variations (occurring above the Nyquist frequency/wavenumber) that comprise in observational datasets spatially uncorrelated noise, using the terminology of optimal interpolation.

We have added following paragraph after line 487:
"We have tested the EOF-based DA in centred time window of 30 days, based mainly on available FerryBox data during the study period. As shown by reconstruction experiments by Elken et al. (2019), the time-dependent method can also work with backward observations as if it occurs during operational forecasts. When more observations become available, for example from new automated buoy stations, Argo floats and gliders, the time window can be shortened. Full covariance matrix estimated from the model results is the backbone of the EOF DA method. Prior and/or complementary to implementation of the method into operational practice, detailed covariance studies using results from multiple models could be useful, as well as additional reconstruction and DA studies using more data sources over longer periods."

- **The previous point is related to a discussion of the main model error sources, which is missing as well**

The model results, accuracy and error problems have been considered by the larger CMEMS community. Unfortunately, references were missing in the model description part, although they were in other places (Golbeck et al., 2015; Hernandez et al., 2015; Tuomi et al., 2018; Huess, 2020; She et al., 2020). They have been added in the section 2.1 of the revised MS. Text on lines 92 is extended to:

"Detailed description of the HBM model and its validation can be found by Berg and Poulsen (2012); further analysis and evaluations are given by Golbeck et al., 2015; Hernandez et al., 2015; Tuomi et al., 2018; Huess, 2020; She et al., 2020. In particular, the CMEMS Quality Information Document (Golbeck et al., 2018) concludes that temperature forecast between the surface and about 100 m depth is one of the major strengths of the CMEMS-V4 product, below the halocline deviations of forecast from observations increase. Regarding salinity, the values are slightly underestimated and the underestimation increases with depth."

- **Observation errors are not discussed at all – this needs to be justified and discussed**

Indeed, this important question was missing in our presentation. In meteorological terminology, our method is "analysis nudging" (e.g. Stauffer and Seaman, 1990) that makes Newtonian relaxation to the gridded fields reconstructed from the observations. The issues of observation errors are included in the reconstruction procedure, when values over (usually very small) sensor space are converted to the values over larger grid cells. DA based on the analysis nudging treat observational errors usually by adding appropriate white noise to the input data, before producing the gridded field to be used in relaxation. In this context, we think we have to make additional study on EOF reconstruction of noisy observations, in order to extend the first results presented by Elken et al. (2019). In this MS, which main focus is on computationally extensive model runs, we add several notes on the problem of observation errors.

Text on lines 164-166 is extended to:

[revised manuscript text omitted]

END

We copy here as an example one test figure (Fig. x2), that was not included in the revised MS, since it has not yet proved to be enough general.

[Figure]

**Figure x2: Example maps of reconstructing SSS based on full covariance matrix using EOF (a) and optimal interpolation (b), and optimal interpolation with Gaussian approximation of covariance, with spatial scale of 150 km.**

- **We missed some discussion on the potential of the method to improve forecasts**

We have added following paragraph after line 487:
"We have tested the EOF-based DA in centred time window of 30 days, based mainly on available FerryBox data during the study period. As shown by reconstruction experiments by Elken et al. (2019), the time-dependent method can also work with backward observations as if it occurs during operational forecasts. When more observations become available, for example from new automated buoy stations, Argo floats and gliders, the time window can be shortened. Full covariance matrix estimated from the model results is the backbone of the EOF DA method. Prior and/or complementary to implementation of the method into operational practice, detailed covariance studies using results from multiple models could be useful, as well as additional reconstruction and DA studies using more data sources over longer periods."

- **We would furthermore appreciate some discussion about the implications of the assimilation on the model dynamics (e.g., vertical density structure, in particular stability)**

We have rewritten lines 497-499:
"There are obvious possible extensions of the EOF DA method to other variables and layers: improvement of stratification modelling, extension to biogeochemical models and DA of oxygen,

nitrogen and phosphorus. Applicability depends on how well the model reproduces the studied fields and their covariance, and much variance is explained by the major EOF modes."

- **It is not really clear, why the authors did not apply a more standard technique, with a more solid theoretical basis. A straightforward approach would be to use a low rank model error covariance matrix based on the presented EOF decomposition in the standard Kalman analysis equation. This would then also include observation errors and avoid the two steps required in the presented technique.**

We have added new section 3.1.1 Covariance, modes and reconstruction tests, given above. Using a full covariance matrix, optimal interpolation of the background field produced in several test similar results to the EOF reconstruction, but these relations need further studies.

Our results indicate that due to the imperfect observational network, model error covariance should also be treated by full covariance matrix. Approximated covariance was found to create too much distortion of the studied fields. Due to taking differences, the error covariance matrix could be more dependent on the model features than the background covariance matrix estimated from the validated model results. Because of absence of model error covariance estimates, we omitted the proposed option in the present study.

**The presentation of the material should be improved. There are deficiencies, in particular with regard to putting the study into context of existing methods and motivating the selected approach. If "computational effort" is the main point, then this has to be quantified better.**

Computational benefits are more elaborated. The paragraph on lines 231-235 is rewritten:

"The above DA method is computationally efficient. The EOF modes are calculated prior to DA cycles. For each DA time step, only one system of linear equations of rank of the number of EOF modes (about 3-6) has to be solved for the entire grid. The coefficients of the matrix are found by summation of the products of EOF mode values over the observation points (Eq. 2). For comparison, optimal interpolation requires solving the system of linear equations of rank of the number of observation points (about 100) for each grid cell (about 1000), with a single inverse matrix calculated for the time step."

**There are quite a view grammar problems and a native speaker should proofread the text.**

We plan additional language check.

**We recommend publication after major revisions.**

We have made substantial revision, added a new subsection and a new figure.

**Specific Comments**

**Abstract**

**We think it would be better to structure the abstract such, that more general information (what is done?) comes first and specific results follow after that**

When preparing the MS, the authors discussed both the options – your proposal and the one we have selected to present. Our choice is based on the better outreach possibilities, as we think.

**Please explain acronym RMSD**

Corrected

**I think it is more common to say "dominating EOF modes" instead of "gravest", but that should be checked by a native speaker**

Corrected.

**Introduction**

**Page 1, line 22: please reformulate "discrepancies of"**

Changed to "errors of".

**Page 2, line 34: replace "then" by "the"**

Corrected. Also, the first word of the sentence is replaced to "Whereas" (formerly "While").

**Data and methods**

**Page 3, line 72: "whichever" instead of "which"**

Corrected.

**Page 3, line 79: "… from the halocline …" please reformulate**

The sentence has been reformulated: "... therefore deeper more saline waters from the halocline of the Baltic Proper penetrate into the Gulf of Finland and form an estuarine halocline also there"

**Page 3, line 87: "better grid cells" instead of "points"**
**It would be good to learn more about the vertical discretization of the model, e.g., how thick is the surface layer.**

Corrected. The end of the sentence is modified "...71 986 of them on the surface with a layer thickness of 3 m".

**Page 4, line 95, maybe better "grid resolution" instead of "grid step"**

Corrected.

**Fig. 2a: Please change color of FerryBox tracks – it cannot be distinguished from land.**
**Please add information on the water depth the FerryBox observations are usually taken.**

We have changed the color of land. We have added new sentences in the Sect 2.2, line 123:

"The analysed water is strongly mixed in the surface layer by the moving ship. Typical observation depth may be considered 5 m, although variations between the ships and due to the variable shipload exist (Lips et al., 2008; Karlson et al., 2016)."

**Page 5: It was not clear, how you interpolate the FerryBox data to a 2D grid. Please explain in more detail.**

The sentence was modified to:

"Two sets of compressed (averaged) FerryBox data were created for further data analysis, containing mean observed values, coordinates and observation times over the selected intervals."

**Page 5, line 142: Did you mean "… too irregular …" ?**

Corrected.

**Page 5, lines 146-150: this paragraph is hardly understandable – please reformulate.**

The paragraph has been reformulated:

"The basic option of EOF reconstruction uses at each DA time step time-fixed amplitudes, encountering the observations spanning over certain time (which can be longer than DA time step) that are transferred to the fixed times by some interpolation or filtering/averaging procedure. The amplitudes are estimated  using time-fixed observations by minimizing the root-mean-square-difference  between the observations and the EOF reconstruction. The amplitudes at adjacent time moments are not directly related, but in case of longer temporal filters when observations overlap  on different DA time steps, indirect relations between adjacent amplitudes become evident."

**Page 6, line 160: "the" instead of "then"**

Corrected.

**Section 2.3**

**The entire section is unfortunately quite messy and confusing, although (as far as I understand) the method is quite basic. The authors have to explain all symbols and indices with much more care. Also, what is a vector or matrix (what size?) and what is a scalar?**

We have used the widespread notation that matrices and vectors are given in upright capital and lowercase bold letters, respectively, and scalars (including elements of matrices and vectors) are given in italic letters. There are two basic sizes of arrays, number of model grid points and number of observations. The presentation is a condensed version of subchapters "Notations for Empirical Orthogonal Functions (EOF)", "Reconstruction of Observed Fields Using EOF Modes" and "Extension of the EOF Reconstruction Method to Time-Dependent Data" by Elken et al. (2019). Although the beginning of Section 2.3 says "...we chose to use EOF reconstruction of large-scale SST and SSS fields, using the orthogonal patterns from models following the detailed outline by Elken et al. (2019)", we checked once more the clarity of condensed material and have rewritten the lines 156-159:

"The main steps of EOF reconstruction are the following. During the standard EOF decomposition, the orthonormal eigenvector matrix $\mathbf{E}$ (contains the spatial eigenvectors $\mathbf{e}_k$) is found from the eigenvalue problem $\mathbf{BE} = \mathbf{\Lambda E}$ , where $\mathbf{B}$ is $M \times M$ spatial covariance matrix, calculated from the $M \times N$ spatio-temporal matrix $\mathbf{X}$ of the "values of interest" by time averaging, and $\mathbf{\Lambda}$ is a diagonal matrix that contains eigenvalues $\lambda_k$."

**Page 6, line 185: "that that"**

Corrected.

**The basic assumption, if you use EOFs for interpolations like you do, is that the covariance structure of the model is correct – this should be stated more explicitly and discussed a little.**

Problems of EOF reconstruction have been considered by Elken et al. (2019). We cite: SST and SSS results are rather well validated by observations and the model-based covariance patterns can be considered trustful. /// Fu et al. (2011) compared covariance patterns from modeled SST and satellite SST, and found them agreeing well. CMEMS QUID report has presented validation of SSS against FerryBox data, showing that the SSS patterns were well simulated by the model. In deeper layers, however, there is usually a larger spread between different model results.

Main differences between actual and model-based covariance estimates are expected within very short term variations (occurring above the Nyquist frequency/wavenumber) that comprise in observational datasets spatially uncorrelated noise, using the terminology of optimal interpolation.

We have added on line 166:
"If white noise with a variance $\varepsilon^2$ is present in the decomposed data due to sub-grid scale processes and/or sampling errors, the noise variance appears only as additive to the diagonal elements of the covariance matrix. The eigenvalue problem becomes $(\mathbf{B} + \varepsilon^2\mathbf{I})\mathbf{E} = \mathbf{\Lambda}\mathbf{E}$, where $\mathbf{I}$ is a unity matrix. Patterns of spatial modes remain unaffected by adding the white noise, but the eigenvalues and energy share of the modes decrease according to a factor $(1 + \varepsilon^2/\sigma^2)^{-1}$. When the sum of eigenvalues of the included dominating modes is less than $\sigma^2 - \varepsilon^2$, contribution of noise is effectively smoothed."

**You could have included observation errors in this interpolation exercise. I guess your assumption at the moment is, that the observations are 100% correct ? - please comment**

Observation errors are considered in the revised text as follows.
Line 217
"This is the main DA calibration parameter, since extensive use of covariance statistics, including the effects of observation errors, has been included in the estimation of gridded reconstruction of point observations."

In the new sub-section 3.1.1
"EOF reconstruction method relies on the full covariance matrix, without any approximation. Full covariance matrix can be implemented in optimal interpolation as well. While EOF method needs to limit the number of included modes, smoothing in such way smaller scale variability and observational errors, optimal interpolation needs to include observational error variance ("nugget effect" in terms of Kriging method, equivalent to optimal interpolation); otherwise the system of underlying linear equations may become close to singular and the result may become unrealistically spiky."

**Eq. 1: you assume that this matrix actually has an inverse – please comment. If the matrix is close to singular, you run into numerical problems as well.**

Eigenvector matrix $\mathbf{E}$ is non-singular, since it is derived from the symmetric covariance matrix $\mathbf{B}$ on the basis of eigenvalue problem $\mathbf{B}\mathbf{E} = \mathbf{\Lambda}\mathbf{E}$. Inclusion of observation operator $\mathbf{H}_i$ ($i$ is the assimilation time index) does not make the determinant of $\mathbf{E}^\mathbf{T}\mathbf{H}_i^\mathbf{T}\mathbf{H}_i\mathbf{E}$ equal to zero, if the number of observations is greater than zero. We excluded the situations with less than 6 observations and singularity was not detected. The cases with too large amplitudes were omitted and DA was not performed (see the text on lines 269-272).

**Section 2.4**

**I guess eq. 1 is a continuous equation, which in its original form should be solved using the internal model time. I assume that you get eq. 4, if you replace the model time step by the assimilation time step – please explain more**

Relaxation by Eq. (3) causes the model state to exponentially approach to the reconstructed grid (target) maps of observations $\psi^o$. If the restoring time scale $\tau$ is much longer than the model time step and still longer than the assimilation time step $\Delta t$, then it is sufficient to apply Eq. (4) with $\Delta t$.

**I had problems to figure out how big the assimilation time step in the experiments actually is – please use consistent notation for critical parameters (e.g. time steps) throughout the document.**

We admit that the notation $\Delta t$, with different indexes, has been used in the first version of the MS in too "distant" contexts - $\Delta t_p = t_p - t_i$ was the difference between the observation and reference times, $\Delta_R$ was the time window and $\Delta t$ was the DA time step. We replaced the variation in time from $\Delta t_p = t_p - t_i$ to $\delta t_p = t_p - t_i$ and $\Delta_R$ to $t_R$.

The values of $t_R$ and $\Delta t$ were presented by words in the beginning of section 3.2: "...using the time-dependent EOF reconstruction method with a time window of 30 days..." and "Further on, each day DA was made on the fine grid using the procedure Eqs. (3)–(4)." We also added mathematical assignments.

**Page 8: line 223: "The DA method … is analogical …" I don't think this is true in this generality, because it seems you don't consider observation errors at all. – please comment. The resemblance with 4DVAR is remote, because there is no model dynamics included in the minimization of the cost function.**

The whole paragraph has been modified and unclear sentences were removed. The modified paragraph is:

"The DA method is based on the full covariance matrix of irregular pattern, calculated from model results over a sufficiently long period. Covariance is further treated using EOF modes. For the reconstruction procedure, we keep the lowest EOF modes without any approximation, covariance from higher modes is truncated. The large-scale features of the EOF reconstruction and associated DA exclude the possibility of creating spurious "bull-eye" patterns around observation points, that may happen for instance during unfavourable selection of optimal interpolation parameters. Subsequently, our DA method handles the large-scale features and excludes the possibility to assimilate smaller scale features, which can be described by the higher modes. The method of time-dependent amplitudes is able to encounter temporally distributed observations, when estimation of linear rate of change of the EOF amplitudes over the selected interval makes sense. Mesoscale deviations from basin-scale EOF patterns follow well-defined covariance decay with space lag; therefore, they could be treated by optimal interpolation with approximated covariance or similar methods (Elken et al., 2018)."

**Page 8, line 239: "… artificial split …" I don't understand this sentence, because this "split" is a standard approach to validate assimilation techniques.**

The sentence has been deleted in this section.

**Page 7, line 217: " … since extensive use has been made …". This is I guess the critical point. The classical approach in an assimilation filter is to combine observations and the model state using covariance information on model errors at each analysis time step. In your approach there are no covariances of model errors. Instead, you use covariances of the background statistics for the interpolation. If you used a scaled version of the background covariance as a proxy for the model error covariance in a classical filter approach, you would probably end up with similar results, but with a more solid theoretical foundation. Anyway, as pointed out in the general comments, the method has to be put into the context of existing methods in a better way.**

We used indeed the background covariance since validated model results are available. We have found that it has complicated structure, but can be physically well interpreted. Encountering the full covariance structure is very important, as we have shown, also in an example of optimal interpolation with full covariance structure. Covariance of model errors is not known in such details. We are not convinced that there is a simple transformation from background covariance to the model error covariance, since it has to be very model-specific, compared to the more universal estimates from validated model results.

We have added a new subsection 3.1.1 as pointed out earlier.

**Section 3.1**

**Page 9, line 275: This is interesting; why don't you show the EOFS computed in your study?**

These results were presented in detail by Elken et al. (2019). We found that repetition of figures is not necessary in this MS since there is open access to the earlier paper.

**Page 13, line 408: The skill is often defined in relation to a reference run (e.g. the free run). In the case of the standard forecast skill, it is a dimensionless number – please check.**

The paragraph has been rewritten:
"Ocean model performance (e.g. Stow et al., 2009; Golbeck et al., 2015; Placke et al., 2018) is usually evaluated by the differences between the observations and the model results, transferred to the times and locations of observations that they can be directly compared. The overall mean difference (over time and space) is termed bias and the standard deviation of differences at all the observation points is denoted as RMSD (centred root-mean-square difference). The forecast skill is usually non-dimensional, with the RMSD of the studied option (in our case, DA) scaled to reference data (FR in our case)."

**Page 16: "There are obvious extensions … layers …"**
**This is, where it gets interesting, because the vertical structure of different model variables (temperature, salinity, etc) is a particular challenge and your assumption about the correctness of model covariances may become a problem (e.g., if the mixed layer thickness in the model is not correct)**

We have extended the clause on line 498:

"Applicability depends on how well the model reproduces the studied fields and their covariance, and how much variance is explained by the major EOF modes."

**Figure 8: It would be interesting to see the absolute differences between observations and the assimilation run and the same for the free run (these differences should reflect both observation and model errors).**

This is a very interesting idea, but we think that adding more details to the figure will compromise readability too much.

[revised manuscript text omitted]

---

## Author Response (AR3)

**os-2020-43-manuscript-version3.pdf**

**Response to the comments by the Reviewers and marked-up manuscript**

General comment 1: My main issue with paper is that I found it a hard to follow at times. Please revise any parts of the paper where I state that it is unclear.

Response: We appreciate very much pointing to the unclear parts of the MS that we as authors were not able to detect.

General comment 2: Also it wasn't clear whether the fine grid assimilation was used or not in the experiments since much of the discussion in the abstract and paper relates to the EOF assimilation. If the fine grid assimilation is done it would be interesting to know what the impact of it is relative to the EOF assimilation. Were any experiments performed only doing EOF or fine grid assimilation to assess the relative impact of both methods?

Response: EOF reconstruction was done on the coarse grid only. Reconstruction results on the coarse grid were interpolated to the fine grid, the grid used in the prognostic model, and subsequently used in the data assimilation scheme in the relaxation part. It is also possible to make covariance matrix estimates and EOF mode calculations on the fine grid, but since we were interested in the first, most energetic modes only, we skipped those calculations since they would not change the lowest mode patterns but would just increase significantly the amount of computations.

I have some more detailed comments below

Comment 1. Abstract p1 line 12-13. I think it would be useful to indicate the size in km or nautical miles. At first I thought the units were in degrees and this would help to avoid that confusion.

Response: Rechecking the notations, we found that notations like "5' N X 10' E" indeed are not widespread and may be not easy to interpret. In order to keep the description of resolution exact, the arc minutes are frequently used. We rewrote the resolutions into "5 X 10 arc minutes by N and E", whereas approximate nautical miles were also noted during the first appearance.

Comment 2. Abstract p1. You don't mention the fine grid assimilation

Response: Indeed, it was not enough clarified in the abstract. We altered the text.

Comment 3. Introduction p1 line 22-24 unclear. If understand it right you are stating that SST in the models is good even without assimilation and SSS is where assimilation has the best impact.

Response: The paper by Placke et al. (2018) says (citation): all models reproduce temperature much better than salinity. We have simplified our sentence according to that.

Comment 4. Introduction p2 line 30. I would change "the operational forecasts within CMEMS" to "the operational Baltic Sea forecasts within CMEMS" just to be specific as other ocean forecasting regions/systems do assimilate data. This might be a good place to explain why data is not currently assimilated in this system as the explanation would be used to motivate the work you are doing here.

Response: We included this refinement and added a note on ongoing work on implementation of DA. One of our co-authors is active member of CMEMS BALMFC team. We know that DA is on the agenda for about 10 years. However, we do not feel confident to analyse the reasons for the delay and note only the need to implement automated DA system which would be robust, reliable and well validated.

Comment 5. Introduction p2 line 44. It is a challenge true but that is not the reason to do the work instead it is because it would be highly beneficial for a region with sparse observations.

Response: That is very good point and we extended our sentence.

Comment 6. Introduction p2 line 46. Perhaps it is a turn of phrase but it is strange to me to cite a paper where the method is presented as an idea in one paper. Was it described theoretically in this paper? Was it demonstrated practically?

Response: We have deleted reference to Elken et al. (2018) and moved it to the section 2.3 where the reconstruction issues are considered in detail.

Comment 7. Introduction p2 line 50-52. It would be useful to clarify how this work builds on Elken 2018 and Elken 2019 for those who haven't seen the other papers.

Response: Since we are keeping only Elken 2019 here then separation of contributions is not needed. Actually, Elken 2018 is a conference paper that shortly presented EOF modes and other statistics, and presented the least squares algorithm to determine "observational" amplitudes when the number of observations is smaller than the number of grid points. Accuracy of the EOF method was tested using pseudo-observations, i.e. model values were extracted in few "observation" points, observational fields over the entire grid were reconstructed, and then compared to the original model fields. Elken 2019 extended the reconstruction method to the time dependent case when "observational" amplitudes were assumed to change linearly in time.

Comment 8. Sec 2.1 p3 line 69 "yearly Maximum SST exceeds usually 15 deg C" -> "the SST usually exceeds 15 deg C in July or August"

Response: Corrected.

Comment 9. Sec 2.1 p3 line 84. It would be good to state the (approximate) grid size in km or nautical miles here.

Response: Corrected.

Comment 10. Sec 2.2 p5 line 140. State the grid size of the coarse grid in km or nautical miles.

Response: Corrected.

Comment 11. Sec 2.2 p5 line 140. You should state that main benefit of the coarse grid is to save computational costs. And state what the downsides are. You do discuss this later so another option would be to writing something like "see section 2.4 for more details on the advantages and disadvantages of using the coarse grid" but it might be better to group all this discussion together here.

Response: Indeed, it was not clear in this point why the coarse grid was introduced. We added explanations, with reference to the more details coming in section 2.4.

Comment 12. Sec 2.3 p5 line 149-152. I think I'd like a little more detail here. You probably should say here that you summarise the EOF method below (in the same section). Also the discussion here gives the impression that you use only the EOF method, but later you do mention a fine grid adjustment.

Response: We agree that more guidance to the reader is needed. In section 2.3 we summarize the well-known EOF decomposition and present general features of EOF reconstruction as a problem when the number of observations is less than the number of EOF modes (equals to the number of model grid cells).

Comment 13. Sec 2.3 p5-6. An additional point to address is are the EOFs multivariate or are SST and SSS treated separately.

Response: The present implementation as given in section 2.4 is univariate. We have added on p6: "Although in the present study we use the dataset X selection as 2D sub-sets of individual oceanographic fields, applications towards multivariate analysis and/or extending over the 3D physical domain are straightforward."

Comment 14. Sec 2.3 p5-6. It would be worth discussing too why the EOFs are not 3D and only need to reconstruct surface temperature and salinity.

Response: We have added on p6: "Although in the present study we use the dataset X selection as 2D sub-sets of individual oceanographic fields, applications towards multivariate analysis and/or extending over the 3D physical domain are straightforward."

Comment 15. Sec 2.3 p8 lines-244-254. I really had trouble following this paragraph describing the use of the full covariance and truncation of the EOF modes. Perhaps it simply needs rewriting for clarity.

Response: This paragraph partially includes the same aspects as are presented later on lines 291-316 and therefore was deleted here, important elements were merged into this later part.

Comment 16. Sec 3.1.2 p11 line 319. Do you mean "… suggested that only the three gravest modes should be included."

Response: This is correct, we made correction.

Comment 17. Sec 3.1.3 p11 line 340. When there is no reconstruction is there no assimilation? (or just the fine scale assimilation?)

Response: We wrote clarification in the modified sentence "In addition, when the number of observations was less than six, reconstruction was not performed and DA step using Eq. (4) was skipped."

Comment 18. Sec 3.2 p12 line 368-376. So is the fine grid assimilation performed in these experiments? If so what is the relative impact of both methods. If not then this needs clarifying.

Response: We have added the clarifications and reformulated this paragraph. In the two-scale DA approach, observations were reconstructed on the coarse grid. Results were interpolated into the fine grid of the model, and subsequently were used for relaxing the fine scale model results towards basin-scale observational patterns.

Comment 19. Sec 3.2 p12 line 368-376. I'm not sure why you have SST01 and SST02. This suggests the data is different. I'm not insisting but wouldn't it be clearer just use the notation DA01 and DA02.

Response: We initially planned to use many codes for the results, like SST01 and SST02, but later we found too many abbreviations making difficult to read. Unfortunately, some unnecessary abbreviations remained and are deleted now. Thanks for finding these inconsistencies, we corrected.

Comment 20. Sec 3.2 p12 line 368-376. Is the same relaxation applied to the EOF and fine scale assimilation?

Response: Observations are calculated using EOFs on the coarse grid. Then they are interpolated into the fine grid and used
for relaxation of fine grid model results. We have reformulated the paragraph, hopefully it is now more clear.

Comment 21. Sec 3.2.1 p13 line 399. Source of the SST satellite data? (I think you describe it earlier if so refer to that section).

Response: Reference to section 2.2 has been added.

Comment 22. Sec 3.2.2 p14 line 437. Could the failure to correct cross-gulf gradients be caused by using coarse EOFs?

Response: This is an interesting note. We have written the modified sentence as: "This implies that DA of surface observations
tends to better correct the mean values than the cross-gulf gradients, for which 3D circulation (presently not assimilated) has significant impact".

Comment 23. Sec 3.2.2 p15 line 445. You mean there is very little difference between DA01 and DA02? Why might that be?

Response: The sentence was not good and has been rewritten to: "The results of assimilation experiments DA01 and DA02, with relaxation times of 10 and 5 days respectively, were not placed between the free run and the observations proportionally
to the corresponding weights 0.1 and 0.2, as can be seen from Fig. 7."

Comment 24. Sec 3.2.4 p16 Confirm that you are assessing forecast skill (i.e. before the observations are assimilated). It is much easier to improve the analysis skill if you are comparing to observations you have assimilated.

Response: We minimized using the term "skill" throughout the MS and renamed the subsection 3.2.4. to "Evaluation of DA-based forecast performance". We compare the DA results (forecast) with a model option when observations are not assimilated
(FR – free run). For clarification, we have modified our sentence to "The forecast skill is usually non-dimensional, with the RMSD of the studied option (in our case, DA) scaled to reference data (FR in our case) as skill = function of [RMSD(DA,FB) / RMSD(FR,FB)]."

Comment 25. Sec 3.2.4 p16 line 492. Too unspecific. Explain what Hernandez et al 2015 said which is relevant here.

Response: We have extended the sentence to: "Hernandez et al. (2015) who reviewed the problems of performance evaluations
of operational ocean models noted that most available observations are used to adjust models and reduce analysis errors. Therefore, a widespread approach is withholding part of the dataset for statistical quantification of errors."

Comment 26. Sec 3.2.4 p17 line 524 DA is not improving the model I would say it improves the forecasting of SSS.

Response: Corrected.

Comment 27. Sec 3.2.4 p17 line 524 "Still, RMSD to the observations makes 62% of observed standard deviations…" I would
write "Still, the forecast RMSD to the observations is 62% of the observed standard deviations which suggests that there may be further room for improvement."

Response: We agree, the sentence was reformulated.

Comment 28. Sec 4 p18 line 570. Do you do both the EOF and some kind of localized DA (fine scale DA)? If you do you are not reducing the computational effort rather you are increasing it. This may still be worthwhile of course.

Response: The sentence was unfortunately not written in a very clear way. Thank you for pointing to the clarity problem. We rewrote: "Firstly, for assimilation of basin-scale patterns, it can be implemented on a coarse grid and therefore it has small computational effort compared to the localized methods like optimal interpolation etc that should be usually implemented on the model resolution, i.e. on the fine grid."

Comment 29. Sec 4 p19 line 575. I'm not sure what you mean here. What has the orthogonality of the EOF basis vectors got do with it?

Response: Indeed, when doing least squares fitting to find "observational" amplitudes, the EOF modes are not anymore orthogonal when they are mapped to the observation points (although full-grid presentation of modes is orthogonal). We corrected the sentence.

Comment 30. Sec 5 p19 line 595. Again is this only assimilating with EOFs or do you make fine scale corrections? What if you just made the fine scale corrections?

Response: We have assimilated only with EOFs. We have mentioned earlier the possibility of making additional localised fine scale assimilation in the regions and times when mesoscale data are available. We hope that our earlier corrections to your comments made it already clear and we kept the sentence on line 595 as it was.

Comment 31. Figure 1 p27. I don't see any arrows indicating river discharges.

Response: We have corrected the figure legend.

Comment 32. Figure 5 p31. Are the satellite observations in the form of a gridded product (there are no gaps in the data from clouds). You should perhaps also state here that the EOF method is only from in-situ data.

Response: We have corrected the figure legend.

Comment 33. Figs 5 and 6 p31-32. Some of the plots y axis go from 57 N to 57 N!

Response: We have corrected the figure images.

Comment 34. Fig 7/8 p33-34. I would find it a bit less confusing if this was DA01 SST and DA02 SST. In section 3.2 you use SST01 SST02 to describe the assimilated EOF reconstructed fields whereas here it appears to be the model fields after assimilation.

Response: We have corrected the figure images.

Comment 35. Fig 9 p35. I assume that the size of the circle has no additional meaning. In the figure legend the large positive is a circle the same size as all the other categories. Perhaps you could redraw the caption or adjust the figure so the caption and points in the plot have matching sizes.

Response: We have corrected the figure images.

**Data assimilation of sea surface temperature and salinity using basin-scale EOF reconstruction: a feasibility study in the NE Baltic Sea**

Mihhail Zujev[1], Jüri Elken[1], Priidik Lagemaa[1]

[1]Department of Marine Systems, Tallinn University of Technology, Tallinn, EE12618, Estonia

*Correspondence to*: Jüri Elken (juri.elken@taltech.ee)

**Abstract.** The tested data assimilation (DA) method based on EOF (Empirical Orthogonal Functions) reconstruction of observations decreased centred root-mean-square difference (RMSD) of surface temperature (SST) and salinity (SSS) in reference to observations in the NE Baltic Sea by 22% and 34%, respectively, compared to the control run without DA. The method is based on the covariance estimates from long period model data. The amplitudes of the pre-calculated dominating

EOF modes are estimated from point observations using least-squares optimization; the method builds the variables on a regular grid. The study used large number of in situ FerryBox observations along four ship tracks from 1 May to 31 December 2015, and observations from research vessels. Within DA, observations were reconstructed as daily SST and SSS maps on the  coarse grid with a resolution of 5 × 10 arc minutes by N and E (ca 5 nautical miles) and subsequently  were interpolated to the fine grid of the prognostic model with a resolution of  0.5 × 1 arc minutes by N and E (ca 0.5 nautical miles). The fine grid observational fields were used in the DA relaxation scheme with daily interval . DA with EOF reconstruction technique was found feasible for further implementation studies, since: 1) the method that works on the large-scale patterns (mesoscale features are neglected by taking only the gravest EOF modes) improves the high-resolution model performance by comparable or even better degree than in the other published studies, 2)

the method is computationally effective.

**1 Introduction**

In the coastal oceans and marginal seas, basin-scale observation, modelling and forecasting of oceanographic and biogeochemical variables is a continuing challenge. As an example from the Baltic Sea, large-scale nutrient dynamics (Andersen et al., 2017; Savchuk, 2018) controls the level of eutrophication and hypoxia, affected by nutrient loads and changing climate (Meier et al., 2019). Placke et al. (2018) have recently shown by comparison of different models, that temperature is much better reproduced than salinity.  Similar evaluation has been obtained earlier by Golbeck et al. (2015), based on 13 operational models used routinely in the Baltic and North Seas.

Data assimilation (DA) is a key element to improve the model accuracy with respect to observations, both in the operational forecast and the reanalysis context (Martin et al., 2015; Buiza et al., 2018; Moore et al., 2019). DA methods are built upon dynamical models and they are based on some kind of minimization (minimum variance, variational cost function formulation etc.) of modelling errors (Carrassi et al., 2018), using estimated statistical characteristics of the studied variables. Most of the widespread methods (optimal interpolation, 3DVar, 4DVar, various options of the Kalman filter, including their ensemble formulations) use covariance as the basic statistical characteristic. Recent overviews on different DA applications in the Baltic Sea can be found in the papers by Liu and Fu (2018), Zujev and Elken (2018), Goodliff et al. (2019), She et al. (2020). Whereas there are several results from Baltic Sea reanalysis studies available (Axell and Liu, 2016; Liu et al., 2017), the operational Baltic Sea forecasts within CMEMS (Copernicus Marine Environment Monitoring Service) do not presently include DA (Huess, 2020) and there is ongoing work to implement automated DA system which would be robust, reliable and well validated.

[revised manuscript text omitted]
 (5 × 10 arc minutes by N and E, about 5 nautical miles 5' N × 10' E) were created. The main benefit of the coarse grid is to save computational costs while keeping the large-scale patterns well resolved (see Sect. 2.4 for more details on the advantages and disadvantages of using the coarse grid). In this procedure, the initial observations were compressed on the coarse grid by roughly 25 times yielding about 13 k average values for SST and SSS. Within the temporal averaging, it was chosen not to apply any diurnal cycle correction and all the observations at different hours were averaged to the closest midnight.

For the interpretation of model and DA results, meteorological data were taken from the model forcing fields. For the occasional comparison, CMEMS remote sensing SST Level 4 (L4) data were retrieved from the service portfolio http://marine.copernicus.eu/services-portfolio/access-to-products/ as the product SST_BAL_SST_L4_NRT_OBSERVATIONS_010_007_b.

**2.3 Reconstruction of gridded data from point observations**

For the purpose of DA, we chose to use EOF reconstruction of large-scale SST and SSS fields, using the orthogonal patterns from models following the detailed outline by Elken et al. (2019), and subsequent relaxation of gridded observations within the model time-stepping. In order to correct the modelled basin-scale patterns towards observations, tThe spatio-temporal distribution of in-situ data was too irregular to use standard interpolation and filtering algorithms like the Cressman method or optimal interpolation with approximated covariance (see an example from the same region by Zujev and Elken, 2018). In this section, we summarize the well-known EOF decomposition and present general features of EOF reconstruction as a problem when the number of observations is less than the number of EOF modes (equals to the number of model grid cells).

The basic option of EOF reconstruction uses at each DA time step time-fixed amplitudes (Elken et al., 2018), encountering the observations spanning over certain time (which can be longer than DA time step) that are transferred to the fixed times by some interpolation or filtering/averaging procedure. The amplitudes are estimated using time-fixed observations by minimizing the root-mean-square-difference between the observations and the EOF reconstruction. The amplitudes at adjacent time moments are not directly related, but in case of longer temporal filters when observations overlap on different DA time steps, indirect relations between adjacent amplitudes become evident.

Elken at al. (2019) proposed also an advanced method with time-dependent amplitudes. Within this approach, the amplitudes and their time derivatives are estimated together with observations within a selected time interval, in order to find least squares between the observations and EOF reconstruction in the observational framework.

The main steps of EOF reconstruction are the following. During the standard EOF decomposition, the orthonormal eigenvector matrix $\mathbf{E}$ (contains the spatial eigenvectors $\mathbf{e}_k$) is found from the eigenvalue problem $\mathbf{BE} = \mathbf{\Lambda E}$ , where $\mathbf{B}$ is $M \times M$ spatial covariance matrix, calculated from the $M \times N$ spatio-temporal matrix $\mathbf{X}$ of the "values of interest" by time averaging, and $\mathbf{\Lambda}$ is a diagonal matrix that contains eigenvalues $\lambda_k$. The dataset $\mathbf{X}$ contains time slices $\mathbf{x}_i$ that are spatial state vectors at time $i$. Although in the present study we use the dataset $\mathbf{X}$ selection as 2D sub-sets of individual oceanographic fields, applications towards multivariate analysis and/or extending over the 3D physical domain are straightforward. 
[revised manuscript text omitted]

**3.2 Data assimilation experiments**

We have used two-scale DA approach (see detailed explanation in Sect. 2.4), where observations were reconstructed on the coarse grid. Results were interpolated into the fine grid of the model, and subsequently were used for relaxing the fine scale model results towards basin-scale observational patterns. More specifically,  gridded observational SST

[revised manuscript text omitted]

Increasing assimilation weight in Eq. (4) two times did not make assimilation results twice closer to the observations. As can be seen from Fig. 7, the results of assimilation experiments DA01 and DA02, with relaxation times of 10 and 5 days respectively, were not placed between the free run and the observations proportionally to the corresponding weights 0.1 and 0.2 . They diverged as the study region experience a temperature drop or daily trend change. Both options of assimilated SST

could either coincide for a long time or go in parallel, but DA02 was systematically closer to the FerryBox observations. Salinity fluctuations had larger amplitudes in the free run without assimilation, but both DA options, with a "thumb" rule - the bigger the weight, the bigger the change, had properly corrected them. Still, in December DA01 showed better results, being closer to the FerryBox salinity than assimilation DA02.

**3.2.3 Spatio-temporal dynamics**

We have chosen to compare assimilation with best results (DA02) to the control run without data assimilation (FR), and track the continuous time-latitude changes of SST and SSS (Fig. 8) in two sub-basins - Gulf of Finland and Gulf of Riga along the coast-to-coast transects given in Fig. 2a. Using DA, temperature was corrected approximately by 1–2 °C, and salinity by less than 1 g kg$^{-1}$. Major systematic change (in the Gulf of Finland this was validated as improvement, see further Sect 3.2.4) was seen near the coasts and in spring/autumn periods, while summer temperatures underwent minor corrections. Salinity corrections had a more uniform distribution and smooth drifting pattern - DA consistently increased SSS values with time and southwards in both of the sub-basins.

[revised manuscript text omitted]

**4 Discussion**

Baltic Sea is considered as one of the most studied marine areas in the world (e.g. Andersen et al., 2017). However, the large observational data sets are distributed unevenly. If we divide our study area into 744 eddy-averaging grid cells of 5 × 10 arc minutes by N and E, then during the study period 330 k FerryBox observations covered only 18% of the sea region. Shipborne monitoring added more 8% coverage of the area, but with much smaller frequency of sampling. Having in mind that the ocean models tend to deviate in the NE Baltic from the observations not only by constant bias but also for large-scale and longer-term response, introduction of non-local, region wide data assimilation is of high importance.

It is interesting to consider how our statistical evaluations of model and DA performance, given in Table 2, compare with other Baltic Sea studies. For remote sensing versus in situ reference, Kozlov et al. (2014) have found RMSD 1.31 °C in the Curonian Lagoon. Uiboupin and Laanemets (2015) have estimated RMSD of various satellite products to FerryBox in the Gulf of Finland from 0.29 to 0.98 °C. Our control run gave RMSD 0.72 °C. Golbeck et al. (2015) compared SST from 13 models with satellite data and found in the Baltic Sea yearly RMSD for SST 0.65–0.87 °C. They found larger relative spread of SSS ensemble members than of SST: deviations in the Gulf of Finland between the models were nearly up to 1 g kg$^{-1}$, while the average SSS is only about 4 g kg$^{-1}$. Unfortunately, there were not enough validating observations for SSS available. Fu et al. (2011) found for the control run RMSD for SST even larger, 1.0 °C, based on satellite observations. They also used DA with ensemble optimal interpolation and found that DA reduced RMSD between the forecasts and observations by 25% for SST and 34% for SSS. With our simpler and less computationally demanding EOF DA technique, similar RMSD reductions have been obtained (Sect. 3.2.4) compared to earlier studies.

We have developed and tested an EOF-based relaxation technique where the large-scale observed fields to be assimilated are pre-calculated independently from the ongoing model. From sparse observations, it is possible to estimate the amplitudes of only the gravest, large-scale EOF modes. The EOF DA method handles large-scale features over the sea basin(s), like change of mean SST, SSS and their gradients, including differential heating in coastal and offshore areas, major patterns from upwelling, and spreading of river discharge. The method can work well with irregular data, but cannot resolve mesoscale features in the areas of dense observations, because the EOF amplitudes of higher modes get noisy, according to our experiments. Optimal interpolation, successive corrections and similar methods assume usually localized covariance and/or radius of influence (e.g. Axell and Liu, 2016); they work well in resolving mesoscale in dense sampling areas, but regions of rare observations remain unaffected by DA. For mesoscale range, in our study area there are only satellite observations of surface variables available. They were omitted from our study, since salinity as a variable of primary interest can be presently determined in the Baltic only in situ. It is possible to implement on top of EOF DA more traditional localized DA methods to assimilate mesoscale data when and where such data are available. Studies on using EOF DA for handling large-scale data are also ongoing in the UK Met Office by Daniel Lea (Haines, 2018).

We have tested the EOF-based DA in centred time window of 30 days, based mainly on available FerryBox data during the study period. As shown by reconstruction experiments by Elken et al. (2019), the time-dependent method can also work with backward observations as if it occurs during operational forecasts. When more observations become available, for example from new automated buoy stations, Argo floats and gliders, the time window can be shortened. Full covariance matrix estimated from the model results is the backbone of the EOF DA method. Prior and/or complementary to implementation of the method into operational practice, detailed covariance studies using results from multiple models could be useful, as well as additional reconstruction and DA studies using more data sources over longer periods.

The EOF DA method has some practical advantages. Firstly, for assimilation of basin-scale patterns, it can be implemented on a coarse grid and therefore 
[revised manuscript text omitted]